# A Formal Framework for Knowledge Acquisition: Going beyond Machine Learning

**DOI:** 10.3390/e24101469

**Published:** 2022-10-14

**Authors:** Ola Hössjer, Daniel Andrés Díaz-Pachón, J. Sunil Rao

**Affiliations:** 1Department of Mathematics, Stockholm University, SE-106 91 Stockholm, Sweden; 2Division of Biostatistics, University of Miami, Miami, FL 33136, USA

**Keywords:** active information, Bayes’ rule, counterfactuals, epistemic probability, learning, justified true belief, knowledge acquisition, replication studies

## Abstract

Philosophers frequently define knowledge as justified, true belief. We built a mathematical framework that makes it possible to define learning (increasing number of true beliefs) and knowledge of an agent in precise ways, by phrasing belief in terms of epistemic probabilities, defined from Bayes’ rule. The degree of true belief is quantified by means of active information $I^{+}$: a comparison between the degree of belief of the agent and a completely ignorant person. Learning has occurred when either the agent’s strength of belief in a true proposition has increased in comparison with the ignorant person ($I^{+}>0$), or the strength of belief in a false proposition has decreased ($I^{+}<0$). Knowledge additionally requires that learning occurs for the right reason, and in this context we introduce a framework of parallel worlds that correspond to parameters of a statistical model. This makes it possible to interpret learning as a hypothesis test for such a model, whereas knowledge acquisition additionally requires estimation of a true world parameter. Our framework of learning and knowledge acquisition is a hybrid between frequentism and Bayesianism. It can be generalized to a sequential setting, where information and data are updated over time. The theory is illustrated using examples of coin tossing, historical and future events, replication of studies, and causal inference. It can also be used to pinpoint shortcomings of machine learning, where typically learning rather than knowledge acquisition is in focus.

## 1. Introduction

### 1.1. The Present Article

The process by which cognitive agents acquire knowledge is complicated, and has been studied from different perspectives within educational science, psychology, neuroscience, cognitive science, and social science [1]. Philosophers usually distinguish between three types of knowledge [2]: acquaintance knowledge (to get to know other persons), knowledge how (to learn certain skills), and knowledge that (to learn about propositions or facts). Mathematically, acquaintance knowledge has been studied via trees and networks, for instance, in small-world-type models and rumor-spreading models [3,4,5]. Knowledge how has been widely developed in education and psychology, since the middle of the twentieth century, by means of testing and psychometry, using classical statistics [6,7,8].

The purpose of this paper is to formulate knowledge that in mathematical terms. Our starting point is to define knowledge that as justified true belief (JTB), which generally is agreed to constitute at least a sufficient condition for such knowledge [9,10]. The primary tools will be the concepts of truth, probabilities, and information theory. Probabilities, in addition to logic, are used to formulate mechanisms of reasoning in order to define beliefs [11,12]. More specifically, a Bayesian approach with subjective probabilities will be used to quantify rational agents’ degrees of beliefs in a proposition. These subjective probabilities may vary between agents, but since each agent is assumed to be rational, its probabilities satisfy basic axioms of probability [13]. This is also referred to as the personalistic view of probabilities in [14].

The degree of belief in a proposition is associated with some type of randomness or uncertainty regarding the truth of the proposition. It is helpful in this context to distinguish between ontological randomness (genuine randomness regarding the truth of the proposition) and epistemic randomness (incomplete knowledge about propositions that are either true or false). Here the focus will be on epistemic randomness, and following [15], subjective probabilities are referred to as epistemic probabilities. The epistemic randomness assumption that each proposition has a fixed truth value can be viewed as a frequentist component of our framework.

To use epistemic probabilities in a wider context of knowledge that (subsequently simply referred to as knowledge), we incorporate degrees of beliefs within a framework of parallel worlds in order to define more clearly what JTB means. These parallel worlds correspond to parameters of a statistical model and a second frequentist notion of one parameter being true, whereas the others are counterfactuals [16]. An agent’s maximal possible discernment between worlds is described in terms of the σ-algebra G. The agent’s degrees of belief are obtained through Bayes’ rule from prior belief and data [17], in such a way that it is not possible to discern between worlds beyond the limits set by G.

Learning is associated with increased degrees of true belief, although these beliefs need not necessarily be justified. More specifically, the agent’s degree of belief in a proposition is compared to that of an ignorant person. This corresponds to an hypothesis test within a frequentist framework. More specifically, the null hypothesis of a proposition being true is tested against an alternative hypothesis that the proposition is false. As a test statistic, we use active information $I^{+}$ [18,19,20], which quantifies how much the agent has learned about the truth value of the proposition compared to an ignorant person. In particular, learning has occurred when the agent’s degree of belief in a true proposition is larger than that of an ignorant person ($I^{+}>0$), or if the agent’s degree of belief in a false proposition is less than that of an ignorant person ($I^{+}<0$). In either case, G sets a limit in terms of the maximal amount of possible learning. Learning is, however, not sufficient for knowledge acquisition, since the latter concept also requires that the true belief is justified, or has been formed for the right reason. Knowledge acquisition is defined as a learning process where the agent’s degree of belief in the true world is increased, corresponding to a more accurate estimate of the true world parameter. Thus, knowledge acquisition goes beyond learning in that it also deals with the "justified" part of the JTB condition. It is related to consistency of a posterior distribution, a notion that is meaningful only within our hybrid frequentist/Bayesian approach.

To the best of our knowledge, the hybrid frequentist/Bayesian approach has only been used in the context of Bayesian asymptotic theory (Section 7.2), but not as a general tool for modeling the distinction between learning and knowledge acquisition. Although the concept of a true world (or the true state of affairs) is used in the context of Bayesian decision theory and its extensions, such as robust Bayesian inference and belief functions based on the Dempster–Shafer theory [21,22,23,24], the goal is then to maximize an expected utility (or to minimize an expected cost) of the agent that makes the decision. In our context, the Bayesian approach is only used to formulate beliefs as posterior distributions, whereas the criteria for learning (probabilities of rejecting a false or true proposition) and knowledge acquisition (consistency) are frequentist. Given that a model with one unique, true world is correct, the frequentist error probability and consistency criteria are objective, since they depend on the true world. No such criteria exist within a purely Bayesian framework.

**Illustration** **1.**
*In order to illustrate our approach for modeling learning and knowledge acquisition, we present an example that will be revisited several times later on. A teacher (the agent) wants to evaluate whether a child has learned addition. The teacher gives the student a home assignment test with two-choice answers, one right and one wrong, to measure the proposition S: “The child is expected to score well on the test." In this case, we have a set $X=\{x_{1},x_{2},x_{3}\}$ of three possible worlds. An ignorant person who does not ask for help is expected to have half her questions right and half her questions wrong ($x_{1}$). A child who knows addition is expected to get a large fraction of the answers right ($x_{2}$). However, there is also a third alternative, where an ignorant student asks for help and is expected to have a high score for that reason ($x_{3}$). Notice in particular that S is true only for the two worlds of the set $A=\{x_{2},x_{3}\}$. If the child answers substantially more questions right than wrong, the active information will be positive and the teacher learns S. However, this learning that S is true does not represent knowledge of whether the student knows how to add, since the teacher is not able to distinguish $x_{2}$ from $x_{3}$. Now, let us say that the test has only two questions. In this setting, it is expected that an ignorant person has one question right and one wrong. However, it is also highly probable that even if the child does not know his sums well, he can answer the two questions in the right way. In this case, the teacher has not learned substantially about S (nor attained knowledge of whether the student knows how to add). The reason is that, since the test has only two questions, the teacher cannot exclude any of $x_{1}$, $x_{2}$, and $x_{3}$. The more questions the test has, and if the student scores well, the more certain the teacher is that either $x_{2}$ or $x_{3}$ is true, that is, the more he learns about S. If the student is also monitored during the exam, alternative $x_{3}$ is excluded and the teacher knows that $x_{2}$ is true; that is, the teacher not only learns about S, but also acquires knowledge that the student knows how to add.*


Each of the following sections contains remarks and illustrations like the previous one. At the end of the paper, a whole section with multiple examples will explore deeper how the model works in practice.

### 1.2. Related Work

Other contributions have been made to developing a mathematical framework for learning and knowledge acquisition. Hopkins [25] studied the theoretical properties of two different models of learning in games, namely, reinforcement learning and stochastic fictitious play. He developed an equivalence relation between the two under a variety of different scenarios with increasing degrees of structure. Stoica and Strack [26] introduced a stochastic model for acquired knowledge and showed that empirical data fit the estimated outcomes of the model well, using data from student performance in university-level classes. Taylor [27] proposed a model using the notion of concept lattices and the mathematical theory of closure spaces to describe knowledge acquisition and organization. However, none of these works has been developed through basic concepts in probability and information theory the way we do here. Our approach permits important generalizations which cover a wide range of real-life scenarios.

## 2. Possible Worlds, Propositions, and Discernment

Consider a collection X of *possible* worlds, of which $x_{0}\in X$ is the true world, and all other worlds $x\in X\;\backslash \;\{x_{0}\}$ are counterfactuals. We will regard *x* as a statistical parameter, and the fact that this parameter has a true but unknown value $x_{0}$ corresponds to a frequentist assumption. The set X is the parameter space of interest, and it is assumed to be either finite or a bounded and open subset of Euclidean space $R^{q}$ of dimension *q*. Let *S* be a proposition (or statement), and impose a second frequentist assumption that *S* is either true or false, although the truth value of *S* may depend on the world x∈X. Define a binary-valued truth function f:X→{0,1} by f(x)=1 or 0, depending on whether *S* is true or not in world *x*. The set A={x∈X;f(x)=1} consists of all worlds for which *S* is a true proposition. Although there is one-to-one correspondence between *f* and *A*, in the sequel it will be convenient to use both notions. The simplest truth scenario of *S* is one for which the truth value of *S* is unique for the true world, i.e.,
(1)$$A_{0}=\left\{\begin{array}{cc} \{x_{0}\}, & if\;f(x_{0})=1,\\ X\;\backslash \;\{x_{0}\}, & if\;f(x_{0})=0. \end{array}\right.$$$x_{0}$ being unique and *f* being binary-valued together correspond to a framework of epistemic randomness, where the actual truth value $f(x_{0})$ of *S* is either 0 or 1. *S* is referred to as *falsifiable* [28] if it is logically possible (in principle) to find a data set *D* implying that the truth value of *S* is 0, or equivalently, that none of the worlds in *A* is true. It is possible though to falsify *S* without knowing $x_{0}$.

## 3. Probabilities

### 3.1. Degrees of Beliefs and Sigma Algebras

Let (X,F) be a measurable space. When X is finite, F consists of all subsets of X (i.e., $F=2^{X}$); otherwise, F is the class of Borel sets. The Bayesian part of our approach is to quantify an agent’s belief in which a world is true by means of an epistemic probability measure *P* on the measurable space (X,F), whereas the beliefs of an ignorant person follow another probability measure $P_{0}$. It is often assumed that
(2)$$P_{0}\left(B\right)=\frac{\left|B\right|}{\left|X\right|},\;\forall B\in F,$$
is the uniform probability measure that maximizes entropy among all probability measures on (X,F), where | · | refers to the cardinality for finite X and to the Lebesgue measure for continuous X. Then, (Equation 2) corresponds to a maximal amount of ignorance about which possible world is true [29]. Sometimes (as in Example 5 below) some general background knowledge is assumed also for the ignorant person, so that $P_{0}$ differs from (Equation 2).

The agent’s and the ignorant person’s strength of belief in *S* are quantified by P(A) and $P_{0}\left(A\right)$, respectively. Following [15], it is helpful to interpret P(A) and $P_{0}\left(A\right)$ as the agent’s and the ignorant person’s predictions of the physical probability $f\left(x_{0}\right)\in \left\{0,1\right\}$ of *S*. Whereas *P* and $P_{0}$ involve epistemic uncertainty, the physical probability is an indicator for the real (physical) event that *S* is true or not.

When an agent’s belief *P* is formed, it is assumed that any information accessible to him, beyond that of the ignorant person, belongs to a sub-σ-algebra G⊂F. This means that the agent has no more knowledge of how to discern events in G than the ignorant person, if this discernment requires that he considers events in F that do not belong to G. Mathematically, this corresponds to a requirement
(3)$$E_{P}\left[g|G^{\prime }\right]=E_{P_{0}}\left[g|G^{\prime }\right],$$
for all F-measurable functions g:X→R, and all sigma algebras $G^{\prime }$ such that $G\subseteq G^{\prime }\subseteq F$. It is assumed, on the left-hand side of (Equation 3), that *g* is a random variable defined on the probability space (X,F,P), whereas *g* is defined on the probability space $\left(X,F,P_{0}\right)$ on the right-hand side of (Equation 3). It follows from (Equation 3) that G sets the limit in terms of the agent’s possibility to form propositions about which world is true. Therefore, G is referred to as the agent’s maximal possible *discernment* about which world is true. It follows from (Equation 3) that
(4)$$\begin{array}{c} \begin{array}{cc} P(A) & =E_{P}\left[f\right]\\ \; & =E_{P}\left\{E_{P}\left[f|G\right]\right\}\\ \; & =E_{P}\left\{E_{P_{0}}\left[f|G\right]\right\}. \end{array} \end{array}$$The minimal amount of discernment corresponds to the trivial σ-algebra $G_{0}=\left\{\emptyset ,X\right\}$. Whenever (Equation 3) holds with $G=G_{0}$, necessarily $P=P_{0}$. This corresponds to removing the outer expectation on the right-hand side of (Equation 4), so that
(5)$$\begin{array}{c} \begin{array}{cc} P_{0}\left(A\right) & =E_{P_{0}}\left[f\right]\\ \; & =E_{P_{0}}\left[f|G_{0}\right]. \end{array} \end{array}$$

**Remark** **1.**
*Suppose there exists an oracle or omniscient agent O that is able to discern between all possible worlds and also knows $x_{0}$. Mathematically, the discernment requirement means that O has knowledge about all sets in a σ-algebra F that corresponds to a maximal amount of discernment between possible worlds. We will assume that f is measurable with respect to F, so that A is measurable (i.e., A∈F). Knowledge of F is, however, not sufficient for knowing A, since A may involve $x_{0}$, as in *(Equation 1)*. By this we mean that if the agent knows F, and if A involves $x_{0}$, then there are several candidates of A for the agent, and he does not know a priori which one of these candidates is the actual A. However, since O knows F and $x_{0}$, he also knows A. It follows that O knows that S is true for all worlds in (the actual) A, and that S is false for all worlds outside of (the actual) A. That is, the oracle knows for which possible worlds the proposition S is true.*


As mentioned in Remark 1, the truth function *f* is measurable with respect to the maximal σ-algebra F. However, depending on how G is constructed, and whether *A* involves $x_{0}$ or not, the set *A* may or may not be known to the agent. Therefore, when *A* involves $x_{0}$, the agent may not be able to compute $P_{0}\left(A\right)$ and P(A) himself. Although he is able to compute $P_{0}\left(B\right)$ and P(B) for all B∈F, since he does not know $x_{0}$, it follows that he does not know which of these sets *B* equals *A*. Therefore, he does not know P(A) and $P_{0}\left(A\right)$, unless P(B)=P(A) and $P_{0}\left(B\right)=P_{0}\left(B\right)$, respectively, for all *B* that are among the agent’s candidates for the set *A*. For instance, suppose X={1,2,3}, P(1)=1/5, P(2)=P(3)=2/5, and A={3}. If the agent’s candidates for *A* are {1}, {2}, and {3}, then the agent does not know P(A). On the other hand, if the agent’s candidates for *A* are {2} and {3}, then he knows P(A), although he does not know *A*.

As will be seen from the examples of Section 8, it is often helpful (but not necessary) to construct G as the σ-algebra that is generated by a random variable *Y* whose domain is X (i.e., G=σ(Y)). This means that *Y* determines the collection G of subsets of X for which the agent is free to form beliefs beyond that of the ignorant person. Typically, *Y* highlights the way in which information is lost by going from F to G. For instance, suppose X=[0,∞) and Y:[0,∞)→{0,1,2,…} is defined by Y(x)=[x/δ] for some δ>0; then, G=σ({[0,δ),[δ,2δ),…}) is the sigma-algebra obtained by from a quantization procedure with accuracy δ.

### 3.2. Bayes’ Rule and Posterior Probabilities

A Bayesian approach will be used to define the agent’s degree of belief *P*. To this end, we regard x∈X as a parameter of a statistical model and that the agent has access to data d∈D. The agent assumes that (x,d) is an observation of a random variable (X,D):Ω→X×D defined on some sample space Ω. The joint distribution of the parameter *X* and data *D*, according to the agent’s beliefs, is $dQ\left(x,D\right)=dP_{0}\left(x\right)L\left(D|x\right)dD$. This is a probability measure on subsets of X×D, with prior distribution $P_{0}$ of the parameter *X*, and with a conditional distribution L(D|x)=dQ(D|x)/dD that corresponds to the likelihood of data *D*. A posterior probability
(6)$$P\left(A\right)=Q\left(A|D\right)=\int_{A}dQ\left(x|D\right)$$
of *A* is formed by updating the prior distribution $P_{0}$ based on data *D*. It is assumed that the likelihood x→L(D∣x) is measurable with respect to G, so that data conform with the agent’s maximal possible discernment between possible worlds. The likelihood function x→L(D∣x) includes the agent’s *interpretation* of *D*. Although this interpretation may involve a subjective part, it is still assumed that the agent is not willing to speculate about possible worlds beyond the limits set by G. That is, whenever the agent discerns events in G beyond the limits set by G, this discernment is the same as for an ignorant person.

**Remark** **2.**
*To account for the possibility that the agent still speculates beyond the limits set by external data, $G=\sigma (G_{\mathrm{ext}},G_{\mathrm{int}})$ could be defined as the smallest σ-algebra containing the σ-algebras $G_{\mathrm{ext}}$ and $G_{\mathrm{int}}$ that originate from external data $D_{\mathrm{ext}}$ and internal data $D_{\mathrm{int}}$ (the agent’s internal experiences, such as dreams and revelations, respectively). Note, however, that x→L(D∣x) is subjective, even when internal data are absent, since agents might interpret external data in different ways, due to the way in which they perceive such data and incorporate previous life experience.*


From Bayes’ rule we find that the posterior distribution satisfies
(7)$$\begin{array}{cc} P(A) & =Q(A|D)\\ \; & =\int_{A}dQ\left(x|D\right)\\ \; & =\frac{L\left(D|A\right)P_{0}\left(A\right)}{L(D)}\\ \; & =\frac{\int_{A}L\left(D|x\right)dP_{0}\left(x\right)}{\int_{X}L\left(D|x\right)dP_{0}\left(x\right)}. \end{array}$$

A couple of additional reasons reinforce the subjectivity of *P*: the prior $P_{0}$ might be subjective, and acquisition of data *D* might vary between agents [30]. Additionally, acquisition of data *D* will not necessarily make *P* more concentrated around the true world $x_{0}$, since it is possible that the data themselves are biased or that the agent interprets the data in a sub-optimal way.

Since the likelihood function is measurable with respect to G, it follows from (Equation 4) that the agent’s belief *P*, after having observed *D*, does not lead to a different discernment between possible worlds beyond G than for an ignorant person. Given G, together with an unlimited amount of unbiased data that the agent interprets correctly, the *G-optimal choice of P* is
(8)$$P\left(B\right)=1\left(x_{0}\in B\right),\;\forall B\in G.$$

Equations (Equation 4) and (Equation 8) uniquely define the G-optimal choice of *P*. Whenever G⊂F is a proper subset of the maximal σ-algebra F, the measure *P* in (Equation 8) is not the same thing as a point mass $\delta_{x_{0}}$ at $x_{0}$. On the other hand, for an oracle with a maximal amount of knowledge about which world is true, G=F, (Equation 8) reduces to a point mass at the true world—i.e.,
(9)$$P=\delta_{x_{0}}⟺P\left(B\right)=1\left(x_{0}\in B\right),\;\forall B\in F.$$

**Remark** **3.**
*An extreme example of biased beliefs is a*
*true-world-excluding probability measure*
*, with support that does not include $x_{0}$:*

(10)
$$supp\left(P\right)\subset X\;\backslash \;\left\{x_{0}\right\}.$$

*Another example is a*
*correct-proposition-excluding probability measure*
*, with support that excludes all worlds x with a correct value $f\left(x\right)=f\left(x_{0}\right)$ of S:*

(11)
$$supp\left(P\right)\subset \left\{\begin{array}{cc} A^{c}=X\;\backslash \;A, & x_{0}\in A,\\ A, & x_{0}\notin A. \end{array}\right.$$



**Illustration** **2**(Continuation of Illustration 1). *Suppose data D∈D={0,1,…,10} are available to the teacher (the agent) in terms of the number of correct answers of a home assignment test with 10 questions. The prior $P_{0}\left(x_{i}\right)=1/3$ is uniform on $X=\{x_{1},x_{2},x_{3}\}$, whereas data $D|x_{i}∼Bin\left(10,\pi_{i}\right)$ have a binomial distribution with probabilities $\pi_{1}=0.5$ and $\pi_{2}=\pi_{3}=0.8$ of answering each question correctly, for a student that either guesses or has math skills/asks for help. Let d be the observed value of D. Since data have the same likelihood ($L\left(d|x_{2}\right)=L\left(d|x_{3}\right)$) for a student who scores well, regardless of whether he knows how to add or gets help, it is clear that the posterior distribution*
P(xi)=L(d|xi)∑j=13L(d|xj)=10dπid(1−πi)10−d∑j=1310dπjd(1−πj)10−d*satisfies $P\left(x_{2}\right)=P\left(x_{3}\right)$. Since the teacher cannot distinguish $x_{2}$ from $x_{3}$, his sigma-algebra*
(12)$$G=\{\emptyset ,\left\{x_{1}\right\},\left\{x_{2},x_{3}\right\},X\}$$*has only four elements, whereas the full sigma-algebra $F=2^{X}$ consists of all eight subsets of X. Note that Equation *(Equation 3)* stipulates that the teacher cannot discern between the elements of X, beyond the limits set by G, better than the ignorant person. In order to verify *(Equation 3)*, since there is no sigma-algebra between G and F, we only need to check this equation for $G^{\prime }=G$. To this end, let g:X→R be a real-valued function. Then, since $P\left(x_{2}\right)=P\left(x_{3}\right)$, it follows that*
$$E_{P}\left(g|G\right)\left(x_{i}\right)=E_{P_{0}}\left(g|G\right)\left(x_{i}\right)=\left\{\begin{array}{cc} g(x_{1}), & i=1,\\ (g\left(x_{2}\right)+g\left(x_{3}\right))/2, & i=2,3. \end{array}\right.$$*in agreement with *(Equation 3)*.*

**Illustration** **3.**
*During the Russo-Japanese war, the czar Nicholas II was convinced that Russia would easily defeat Japan [31]. His own biases (he considered the Japanese weak and the Russians superior) and the partial information he received from his advisors blinded him to reality. In the end, Russian forces were heavily beaten by the Japanese. In this scenario, the proposition S is “Russia will beat Japan", X consists of all possible future scenarios, and f(x)=1 for those scenarios x∈X in which Russia would win the war. As history reveals, $f(x_{0})=0$. The information he received from his advisors was D, and we know it was heavily biased. Nicholas II adopted (very subjectively!) a correct-proposition-excluding probability measure, as in *(Equation 11)*, because he did not even consider the possibility of Russia being defeated. The main reason was a dramatically poor assessment of the likelihood L(D∣x), for x∈X, on top of a prior $P_{0}$ that had a low probability for scenarios $x\in A^{c}$. Nicholas II’s verdict was P(A)≈1.*


### 3.3. Expected Posterior Beliefs

Since *D* is random, so is *P*. For this reason, the expected posterior distribution
(13)$$\bar{P}\left(B\right)=E_{x_{0}}\left[Q\left(B|D\right)\right]=E_{x_{0}}\left[P\left(B\right)\right],\;\forall B\in F,$$
will be used occasionally, with an expectation corresponding to averaging over all possible data sets *D* according to its distribution $L(\cdot |x_{0})$ in the true world. Consequently, $\bar{P}\left(A\right)$ represents the agent’s expected belief in *S* in the true world $x_{0}$. Note in particular that in contrast to the posterior *P*, the expected posterior $\bar{P}$ is not a purely Bayesian notion, since it depends on $x_{0}$.

## 4. Learning

### 4.1. Active Information for Quantifying the Amount of Learning

The active information (AIN) of an event *B* is
(14)$$I^{+}\left(B\right)=\log\frac{P(B)}{P_{0}\left(B\right)}.$$In particular, $I^{+}\left(A\right)$ quantifies how much an agent has learned about whether *S* is true or not compared to an ignorant person. By inserting (Equation 7) into (Equation 14), we find that the AIN
(15)$$I^{+}\left(A\right)=\log\frac{L\left(D|A\right)}{L(D)}=\log\frac{\int L\left(D|x\right)dP_{0}\left(x|A\right)}{\int L\left(D|x\right)dP_{0}\left(x\right)}$$
is the logarithm of the ratio between how likely it is to observe data when *S* holds, and how likely data are when no assumption regarding *S* is made (see also [32]). The corresponding AIN for expected degrees of beliefs is
(16)$${\bar{I}}^{+}\left(A\right)=\log\frac{\bar{P}\left(A\right)}{P_{0}\left(A\right)}.$$

**Definition** **1**(Learning). *Learning about S has occurred (conditionally on observed D) if the probability measure P either satisfies $I^{+}\left(A\right)>0$ when $x_{0}\in A$ or $I^{+}\left(A\right)<0$ when $x_{0}\notin A$. In particular, full learning corresponds to $I^{+}\left(A\right)=-\log P_{0}\left(A\right)$ when $x_{0}\in A$ and $I^{+}\left(A\right)=-\infty$ when $x_{0}\notin A$. Learning is expected to occur if the probability measure $\bar{P}$ is such that ${\bar{I}}^{+}\left(A\right)>0$ when $x_{0}\in A$ or ${\bar{I}}^{+}\left(A\right)<0$ when $x_{0}\notin A$. In particular, full learning is expected to occur if ${\bar{I}}^{+}\left(A\right)=-\log P_{0}\left(A\right)$ when $x_{0}\in A$ or ${\bar{I}}^{+}\left(A\right)=-\infty$ when $x_{0}\notin A$.*

**Remark** **4.**
*Two extreme scenarios for the active information, when $x_{0}\in A$, are*

(17)
$$I^{+}\left(A\right)\overset{x_{0}\in A}{=}\left\{\begin{array}{cc} -\log P_{0}\left(A\right), & \mathrm{if}\;(8)\;\mathrm{holds}\;\mathrm{and}\;A\in G,\\ -\infty , & \mathrm{if}\;(11)\;\mathrm{holds}. \end{array}\right.$$


*According to Definition 1, the upper part of *(Equation 17)* represents full learning—that is, P(A)=1; whereas the lower part corresponds to a maximal amount of false belief about S when $x_{0}\in A$—that is, P(A)=0.*


**Remark** **5.**
*Suppose S is a proposition that a certain entity or machine functions; then, $-\log P_{0}\left(A\right)$ is the functional information associated with the event A of observing such functioning entity [33,34,35]. In our context, functional information corresponds to the maximal amount of learning about S when the machine works ($f(x_{0})=1$).*


### 4.2. Learning as Hypothesis Testing

It is possible to view the AIN in (Equation 15) as a test statistic for choosing between the two statistical hypotheses
(18)$$\begin{array}{cc} H_{0}: & S\;\mathrm{is true}⟺x_{0}\in A,\\ H_{1}: & S\;\mathrm{is false}⟺x_{0}\notin A, \end{array}$$
with the null distribution $H_{0}$ being rejected (conditionally on observed *D*) when
(19)$$I^{+}\left(A\right)\leq I$$
for some threshold *I* [36,37,38]. Typically, this threshold represents a lower bound of what is considered to be a significant amount of learning when *S* is true. Note in particular that the framework of the hypothesis test, (Equation 18) and (Equation 19), is frequentist, although we use Bayesian tools (the prior and posterior distributions) to define the test statistic.

In order to introduce performance measures of how much the agent has learnt, let ${\mathrm{Pr}}_{x_{0}}$ refer to a probabilities when data $D∼L(\cdot |x_{0})$ are generated according to what one expects in the true world. The type I and II errors of the test (Equation 18) and (Equation 19) are then defined as
(20)$$\begin{array}{cccc} \alpha (x_{0}) & = & {\mathrm{Pr}}_{x_{0}}\left[I^{+}\left(A\right)\leq I\right], & x_{0}\in A,\\ \beta (x_{0}) & = & {\mathrm{Pr}}_{x_{0}}\left[I^{+}\left(A\right)>I\right], & x_{0}\notin A, \end{array}$$
respectively. Both these error probabilities are functions of $x_{0}$, and they quantify how much the agent has learnt about the truth (cf. Figure 1 for an illustration).

### 4.3. The Bayesian Approach to Learning

Within a Bayesian framework, we think of $H_{0}$ and $H_{1}$ as two different models, *A* and $A^{c}$, that represent a subdivision of the parameter space into two disjoints subsets. The posterior odds
(21)$$\mathrm{PostOdds}=\frac{1-P(A)}{P(A)}=\frac{1-P_{0}\left(A\right)}{P_{0}\left(A\right)}\cdot \frac{L(D|A^{c})}{L(D|A)}=\frac{1-P_{0}\left(A\right)}{P_{0}\left(A\right)}\cdot \mathrm{BF}$$
factor into a product of the prior odds and the Bayes factor. Hypothesis $H_{1}$ is chosen whenever
(22)PostoOdds≥r,
for some threshold *r*. If the cost of drawing a parameter X∼P from *A* ($A^{c}$) is $C_{0}$ ($C_{1}$) when $H_{1}$ ($H_{0}$) is chosen, the optimal Bayesian decision rule corresponds to $r=C_{0}/C_{1}$. A little algebra reveals that the AIN is a monotone decreasing function
$$I^{+}=-\log\left[P_{0}\left(A\right)\left(1+\mathrm{PostoOdds}\right)\right]$$
of the posterior odds. From this, it follows that the frequentist test (Equation 19), with AIN as test statistic, is equivalent to the Bayesian test (Equation 22), whenever $I=-\log[P_{0}\left(A\right)\left(1+r\right)]$. However, the interpretation of the two tests differ. Whereas the aim of the Bayesian decision rule is to minimize an expected cost (or maximize an expected reward/utility), the aim of the frequentist test is to keep the error probabilities of type I and II low.

### 4.4. Test Statistic When $x_{0}$ Is Unknown

Recall that the agent may or may not know the set *A*. In the latter case, the agent cannot determine the value of the test statistic $I^{+}\left(A\right)$, and hence he cannot test between $H_{0}$ and $H_{1}$ himself. This happens, for instance, for the truth function (Equation 1), with $A=\{x_{0}\}$, since the AIN $I^{+}\left(A\right)=\log\left[p\left(x_{0}\right)/p_{0}\left(x_{0}\right)\right]$ then involves the unknown $x_{0}$, with p(x)dx=dP(x) and $p_{0}\left(x\right)dx=dP_{0}\left(x\right)$. Although $I^{+}\left(A\right)$ is not known for this particular choice of *A*, the agent may still use the posterior distribution (Equation 7) in order to compute the expected value (conditionally on observed *D*)
(23)$$\begin{array}{ccc} E_{Q}\left[I^{+}\left(\left\{X\right\}\right)|D\right] & = & E_{Q}\left[\log\frac{p(X)}{p_{0}\left(X\right)}|D\right]\\ & = & E_{P}\left[\log\frac{p(X)}{p_{0}\left(X\right)}\right]\\ & = & \int_{X}\log\frac{p(x)}{p_{0}\left(x\right)}p\left(x\right)dx\\ & = & D_{KL}\left(P∥P_{0}\right)\\ & = & H\left(P,P_{0}\right)-H\left(P\right) \end{array}$$
of the test statistic according to his posterior beliefs. Note that (Equation 23) equals the Kullback–Leibler divergence $D_{KL}\left(P∥P_{0}\right)$ between *P* and $P_{0}$, or the difference between the cross entropy $H(P,P_{0})$ between *P* and $P_{0}$, and the entropy H(P) of *P*. If we also take randomness of the data set *D* into account, and make use of (Equation 7), it follows that the expected AIN, for the same choice of *A*, equals the mutual information
(24)$$\begin{array}{ccc} E_{Q}\left[I^{+}\left(\left\{X\right\}\right)\right] & = & E_{Q}\left[E_{Q}\left[\log\frac{p(X)}{p_{0}\left(X\right)}|D\right]\right]\\ & = & \int \log\frac{L(d|x)}{L(d)}dQ\left(x,d\right)\\ & = & \int \log\frac{q(x,d)}{p_{0}\left(x\right)L\left(d\right)}dQ\left(x,d\right), \end{array}$$
between *X* and *D*, when (X,D)∼Q vary jointly according to the agent’s Bayesian expectations, and with q(x,d)=dQ(x,d)/d(x,d).

## 5. Knowledge Acquisition

### 5.1. Knowledge Acquisition Goes beyond Learning

As mentioned in the introduction, knowledge acquisition goes beyond learning, since it also requires that a true belief in *S* is justified (see Figure 2 for an illustration).

It is possible, in principle, for an agent whose probability measure *P* corresponds to a smaller belief in $x_{0}$ compared to that of the ignorant person, to have a value of $I^{+}$ anywhere in the range $\left[-\infty ,-\log P_{0}\left(A\right)\right]$ when *S* is true (i.e., when $x_{0}\in A$). One can think of a case in which the agent will believe in *S* with certainty (P(A)=1) if supp(P)⊂A; but this belief in *S* is for the wrong reason if, for instance, the agent does not believe in the true world, i.e., if (Equation 10) holds, corresponding to the left part of Figure 2. Another less extreme situation occurs when the agent has a higher belief in *A* compared to the ignorant person but has lost some (but not all) confidence in the true world with respect to that of the ignorant person; in this case, the agent has not acquired new knowledge about the true world compared to the ignorant person, although he still has learned about *S* and has some knowledge about the true world.

### 5.2. A Formal Definition of Knowledge Acquisition

Knowledge acquisition is formulated using tools from statistical estimation theory. Loosely speaking, the agent acquires knowledge, based on data *D*, if the posterior distribution *P* gets more concentrated around $x_{0}$, compared to an *ignorant* person. By this we mean that each closed ball centered at $x_{0}$ has a probability that is at least as large under *P* as under $P_{0}$. Closed balls require, in turn, the concept of a metric or distance; that is, a function d:X×X→[0,∞). Some examples of metric are:If $X\subset R^{q}$, we use the Euclidean distance $d\left(x_{1},x_{2}\right)=\sqrt{\sum_{i=1}^{q}{\left(x_{2i}-x_{1i}\right)}^{2}}$ between $x_{1},x_{2}\in X$ as metric.If $X={\left\{0,1\right\}}^{q}$ consists of all binary sequences of length *q*, then $d\left(x_{1},x_{2}\right)=\sum_{i=1}^{q}\left|x_{2i}-x_{1i}\right|$ is the Hamming distance between $x_{1}$ and $x_{2}$.If X is a finite categorical space, we put
$$d\left(x_{1},x_{2}\right)=\left\{\begin{array}{cc} 0, & x_{1}=x_{2},\\ 1, & x_{1}\neq x_{2}. \end{array}\right.$$

Equipped with a metric on X, knowledge acquisition is now defined:

**Definition** **2**(Knowledge acquisition and full knowledge). *Let $B_{\epsilon }\left(x_{0}\right)=\left\{x\in X:d\left(x,x_{0}\right)\leq \epsilon \right\}$ be the closed ball of radius ϵ that is centered at $x_{0}$ with respect to some metric d. We say that an agent has acquired knowledge about S (conditionally on observed D) if learning has occurred according to Definition 1, and in order for this learning to be justified, the following two properties are satisfied for all ϵ>0:*
(25)$$P\left(B_{\epsilon }\left(x_{0}\right)\right)>0,$$*and*
(26)$$P\left(B_{\epsilon }\left(x_{0}\right)\right)\geq P_{0}\left(B_{\epsilon }\left(x_{0}\right)\right)$$*with strict inequality for at least one ϵ>0. Full knowledge about S requires that *(Equation 9)* holds; i.e., that the agent with certainty believes that the true world $x_{0}$ is true. The agent is expected to acquire knowledge about S if learning is expected to occur, according to Definition 1, and if *(Equation 25)* and *(Equation 26)* hold with 
 is true. The agent is expected to acquire knowledge about S if learning is expected to occur, according to Definition 1, and if *(Equation 25)* and *(Equation 26)* hold with $\bar{P}$ instead of P. The agent is expected to acquire full knowledge about S if *(Equation 9)* holds with $\bar{P}$ instead of P.*

Several remarks are in order.

**Remark** **6.**
*Property *(Equation 25)* ensures that $x_{0}$ is in support of P ([39], p. 20) Kallenberg2021a. When $P_{0}$ is the uniform distribution *(Equation 2)*, (*Equation 25)* follows from *(Equation 26)*. Property *(Equation 26)* is equivalent to $I^{+}\left(B_{\epsilon }\left(x_{0}\right)\right)\geq 0$, when $P_{0}\left(B_{\epsilon }\left(x_{0}\right)\right)>0$. In this case, the requirement that (Equation 26) is satisfied with strict inequality for some $\epsilon =\epsilon^{*}>0$ is equivalent to learning the proposition $S_{\epsilon^{*}}$: "The distance of a world to the true world $x_{0}$ is less than or equal to $\epsilon^{*}$," corresponding to a truth function*

(27)
$$f_{\epsilon^{*}}\left(x\right)=1\left(x\in B_{\epsilon^{*}}\left(x_{0}\right)\right).$$

*Since the agent does not know $x_{0}$, neither $f_{\epsilon^{*}}$ nor $A_{\epsilon^{*}}=\left\{x\in X;\;f_{\epsilon^{*}}\left(x\right)=1\right\}$ is known to him, even if he is able to discern between all possible worlds. If $f_{\epsilon^{*}}$ differs from the original truth function f, learning of $S_{\epsilon^{*}}$ can be viewed as meta-learning. Note also that $A_{0}=\left\{x_{0}\right\}$ corresponds to the set in (Equation 1).*


**Remark** **7.**
*Suppose the truth function used to define learning and knowledge acquisition satisfies *(Equation 27)*, i.e., $f=f_{\epsilon }$ for some ϵ≥0. Then *(Equation 25)* and *(Equation 26)* are sufficient for knowledge acquisition, since they imply that learning of $S=S_{\epsilon^{*}}$ has occurred, according to Definition 1. Although knowledge acquisition in general requires more than learning, the two concepts are equivalent for a truth function $f=f_{0}$, with $A=A_{0}=\left\{x_{0}\right\}$, as defined in *(Equation 1)*. Indeed, in this case it is not possible to learn whether $S=S_{0}$ is true or not for the wrong reason.*


**Remark** **8.**
*Recall from Definition 1 that an agent has fully learnt S when*

(28)
$$P\left(A\right)=1\left(x_{0}\in A\right)=\left\{\begin{array}{cc} 1, & x_{0}\in A,\\ 0, & x_{0}\notin A. \end{array}\right.$$

*For a rational agent, the lower part of *(Equation 28)* should hold when data D falsifies S. In general, *(Equation 28)* is a necessary but not sufficient condition for full knowledge. Indeed, it follows from *(Equation 9)* that, for a person to have full knowledge, $P\left(B\right)=1\left(x_{0}\in B\right)$ must hold for all B∈F, not only for the set A of worlds for which S is true.*


**Remark** **9.**
*Suppose a distance measure d(P,Q) between probability distributions on (X,F) is defined. This gives rise to a different definition of knowledge acquisition, whereby the agent acquires knowledge if has learnt about S and additionally $d\left(P,\delta_{x_{0}}\right)<d\left(P_{0},\delta_{x_{0}}\right)$, that is, if his beliefs are closer than the ignorant person’s beliefs to the Oracle’s beliefs. Possible choices of distances are the Kullback–Leibler divergence $d\left(P,Q\right)=D_{KL}\left(Q||P\right)$ and the Wasserstein metric $d\left(P,Q\right)=\min_{X_{1},X_{2}}E\left|X_{1}-X_{2}\right|$, where the minimum is taken over all random vectors $\left(X_{1},X_{2}\right)$ whose marginals have distributions P and Q, respectively. Note in particular that the KL choice of distance yields $d\left(Q,\delta_{x_{0}}\right)=-\log Q\left(x_{0}\right)$. The corresponding notion of knowledge acquisition is weaker than in Definition 2, requiring *(Equation 25)* and *(Equation 26)* to hold only for ϵ=0.*


**Illustration** **4**(Continuation of Illustration 1). *To check whether learning or knowledge acquisition has occurred, according to Definitions 1 and 2, for the student who takes the math home assignment, $x_{0}$ must be known. The reader may think of an instructor with full information—an F-optimal measure according to *(Equation 9)*—who checks whether a pupil has learned and acquired knowledge or not. However, in Illustration 1 it is the teacher who is the pupil and learns and acquires knowledge about the skills of a math student. In this context, the instructor is a supervisor of the teacher who knows whether the math student is able to add ($x_{0}=x_{2}$) or not, and in the latter case whether the student gets help ($x_{0}=x_{3}$) or not ($x_{0}=x_{1}$). Whereas the instructor’s sigma-algebra is F, the teacher’s sigma-algebra G in *(Equation 12)* does not make it possible to discern between $x_{2}$ and $x_{3}$. Suppose $x_{0}=x_{2}$. No matter how many questions the home exam has, as long as the teacher does not get information from the instructor on whether the student solved the home exam without help or not, although the teacher learns that S is true, since the student scores well, he will never acquire full knowledge that the student knows how to add, since $P\left(x_{0}\right)=P\left(x_{2}\right)=P\left(x_{3}\right)\leq 0.5<1$.*

## 6. Learning and Knowledge Acquisition Processes

The previous two sections dealt with learning and knowledge acquisition of a static belief *P*, corresponding to an agent who is able to discern between worlds according to one sub-σ-algebra G of F, and who has access to one data set *D*. The setting is now extended to consider an agent who is exposed to an increasing amount of information about (or discernment between) the possible worlds in X, and increasingly larger data sets.

### 6.1. The Process of Discernment and Data Collection

Mathematically, an increased ability to discern between possible worlds is expressed as a sequence of σ-algebras
(29)$$G_{1}\subset \ldots \subset G_{n}\subset F.$$Typically, $G_{k}$ is generated by a random variable $Y_{k}$ whose domain is in X for k=1,…,n. The σ-algebras in (Equation 29) are associated with increasingly larger data sets $D_{1},\ldots ,D_{n}$, with $D_{k}\in D_{k}$. Let $dQ_{k}\left(x,D_{k}\right)=dP_{0}\left(x\right)L\left(D_{k}|x\right)dD_{k}$ refer to the joint distribution of the parameter and data in step *k*, such that the likelihood $x\to L\left(D_{k}|x\right)$ of $D_{k}$ is $G_{k}$-measurable. This implies that an agent who interprets data $D_{k}$ according to this likelihood function has beliefs (represented by the posterior probability measure $P_{k}\left(\cdot \right)=Q_{k}\left(\;\cdot |D_{k}\right)$) that correspond to not being able to discern events outside of $G_{k}$ better than an ignorant person. Mathematically, this is phrased as a requirement
(30)$$E_{P_{k}}\left[g|G_{k}^{\prime }\right]=E_{P_{0}}\left[g|G_{k}^{\prime }\right],$$
for all F-measurable functions g:X→R and sigma algebras $G_{k}^{\prime }$ such that $G_{k}\subset G_{k}^{\prime }\subset F$, for k=1,…,n. The collection of pairs $\left(D_{1},G_{1}\right),\ldots ,\left(D_{n},G_{n}\right)$ is referred to as a discernment and data collection process. The active information, after *k* steps of the discernment and data collection process, is
(31)$$I_{k}^{+}\left(A\right)=\log\frac{P_{k}\left(A\right)}{P_{0}\left(A\right)}.$$

Let ${\bar{P}}_{k}\left(\cdot \right)=E_{x_{0}}\left[P_{k}\left(\;\cdot |D_{k}\right)\right]$ refer to expected degrees of belief after *k* steps of the information and data collection process, if data $D_{k}∼L\left(\cdot |x_{0}\right)$ vary according that what one expects in the true world. The corresponding active information is
(32)$${\bar{I}}_{k}^{+}\left(A\right)=\log\frac{{\bar{P}}_{k}\left(A\right)}{P_{0}\left(A\right)}.$$

In the following sections we will use the sequences $I_{1}^{+},\ldots ,I_{n}^{+}$ and $P_{1},\ldots ,P_{n}$ of AINs and posterior beliefs in order to define different notions of learning and knowledge acquisition.

### 6.2. Strong Learning and Knowledge Acquisition

**Definition** **3**(Strong learning). *The probability measures $P_{1},\ldots ,P_{n}$, obtained from the discernment and data collection process represent a learning process in the strong sense (conditionally on observed $D_{1},\ldots ,D_{n}$) if*
(33)$$\left\{\begin{array}{cc} 0\leq I_{1}^{+}\left(A\right)\leq \cdots \leq I_{n}^{+}\left(A\right), & if\;x_{0}\in A,\\ 0\geq I_{1}^{+}\left(A\right)\geq \cdots \geq I_{n}^{+}\left(A\right), & if\;x_{0}\notin A, \end{array}\right.$$*with at least one strict inequality. Learning is expected to occur, in the strong sense, if *(Equation 33)* holds with ${\bar{I}}_{1}^{+}\left(A\right),\ldots ,{\bar{I}}_{n}^{+}\left(A\right)$, instead of $I_{1}^{+}\left(A\right),\ldots ,I_{n}^{+}\left(A\right)$.*

**Definition** **4**(Strong knowledge acquisition). *With $B_{\epsilon }\left(x_{0}\right)$ as in Definition 2, the learning process is knowledge acquiring in the strong sense (conditionally on observed $D_{1},\ldots ,D_{n}$) if, in addition to *(Equation 33)*, we have that this learning process is justified, so that for all ϵ>0, $P_{1}\left(B_{\epsilon }\left(x_{0}\right)\right)>0$ and*
(34)$$P_{0}\left(B_{\epsilon }\left(x_{0}\right)\right)\leq P_{1}\left(B_{\epsilon }\left(x_{0}\right)\right)\leq \cdots \leq P_{n}\left(B_{\epsilon }\left(x_{0}\right)\right),$$*with strict inequality for at least one step of *(Equation 34)* and for at least one ϵ>0. Knowledge acquisition is expected to occur, in the strong sense, if learning is expected to occur in the strong sense, according to Definition 3, and additionally *(Equation 34)* holds with ${\bar{P}}_{1},\ldots ,{\bar{P}}_{n}$, instead of $P_{1},\ldots ,P_{n}$.*

**Illustration** **5**(Continuation of Illustration 1). *Assume the teacher of the math student has a discernment and data collection process $\left(G_{1},D_{1}\right),\left(G_{2},D_{2}\right)$, where in the first step, $G_{1}=G$ and $D_{1}|x_{i}∼Bin\left(10,\pi_{i}\right)$ are obtained from a home assignment with 10 questions (as described in Section 3.2). Suppose the student knows how to add ($x_{0}=x_{2}$). It can be seen that*
(35)$$\begin{array}{ccc} P_{1}\left(A\right) & = & P_{1}\left(x_{2}\right)+P_{1}\left(x_{3}\right)>P_{0}\left(A\right)=2/3,\\ P_{1}\left(x_{0}\right) & = & P_{1}\left(x_{2}\right)>P_{0}\left(x_{2}\right)=1/3 \end{array}$$*whenever $7\leq d_{1}\leq 10$. Assume that in a second step the teacher receives information $Z_{2}\in \left\{0,1\right\}$ from the instructor on whether the student used external help ($Z_{2}=1$) or not ($Z_{2}=0$) during the exam. Let $d_{2}=\left(d_{1},z_{2}\right)$ refer to observed data after step 2. The likelihood, after the second step, then takes the form*
L(d2|xi)=10d1πid1(1−πi)10−di·L(z2|xi),*where $L\left(1|x_{i}\right)=1\left(x_{i}=x_{3}\right)$ and $L\left(0|x_{i}\right)=1\left(x_{i}\in \left\{x_{1},x_{2}\right\}\right)$. If the instructor correctly reports that the student did not use external help ($z_{2}=0$), it follows that*
(36)$$\begin{array}{ccc} P_{2}\left(A\right) & = & P_{2}\left(x_{2}\right)=P_{1}\left(x_{2}\right)/\left(P_{1}\left(x_{1}\right)+P_{1}\left(x_{2}\right)\right)<2P_{1}\left(x_{2}\right)/\left(P_{1}\left(x_{1}\right)+2P_{1}\left(x_{2}\right)\right)=P_{1}\left(A\right),\\ P_{2}\left(x_{0}\right) & = & P_{2}\left(x_{2}\right)>P_{1}\left(x_{2}\right)/\left(P_{1}\left(x_{1}\right)+2P_{1}\left(x_{2}\right)\right)=P_{1}\left(x_{2}\right)=P_{1}\left(x_{0}\right). \end{array}$$*We deduce from *(Equation 35)* and *(Equation 36)* that*
(37)$$P_{0}\left(x_{0}\right)<P_{1}\left(x_{0}\right)<P_{2}\left(x_{0}\right),$$*which suggests that knowledge acquisition has occurred if the categorical space metric $d\left(x_{i},x_{j}\right)=1\left(x_{i}\neq x_{j}\right)$ is used on X. However, since $P_{2}\left(A\right)<P_{1}\left(A\right)$, neither learning nor knowledge acquisition in the strong sense has occurred. The reason is that the information from the instructor (that the student has not cheated) makes the teacher less certain as to whether the student is able to score well on the test. On the hand, if we change the proposition to S: "The student knows how to add," with $A=\{x_{2}\}$, then strong learning and knowledge acquisition has occurred because of *(Equation 37)*, since $P_{k}\left(A\right)=P_{k}\left(x_{0}\right)$ for k=0,1,2.*

### 6.3. Weak Learning and Knowledge Acquisition

Learning and knowledge acquisition are often fluctuating processes, and the requirements of Definition 3 are sometimes too strict. Accordingly, weaker versions of learning and knowledge acquisition are thus introduced.

**Definition** **5**(Weak learning). *Learning in the weak sense has occurred at time n (conditionally on observed $D_{n}$), if*
(38)$$\left\{\begin{array}{cc} 0<I_{n}^{+}\left(A\right), & if\;x_{0}\in A,\\ 0>I_{n}^{+}\left(A\right), & if\;x_{0}\notin A. \end{array}\right.$$*Learning is expected to occur in the weak sense if (Equation 38) holds with ${\bar{I}}_{n}^{+}$ instead of $I_{n}^{+}$.*

**Definition** **6**(Weak knowledge acquisition). *Knowledge acquisition in the weak sense occurs (conditionally on observed $D_{n}$) if, in addition to the weak learning condition *(Equation 38)*, in order for this learning to be justified, it holds for all ϵ>0 that $P_{n}\left(B_{\epsilon }\left(x_{0}\right)\right)>0$ and*
(39)$$P_{0}\left(B_{\epsilon }\left(x_{0}\right)\right)\leq P_{n}\left(B_{\epsilon }\left(x_{0}\right)\right),$$*with strict inequality for at least one ϵ>0. Knowledge acquisition is expected to occur in the weak sense if weak learning occurs according to Definition 5 and *(Equation 39)* holds with ${\bar{P}}_{n}$ instead of $P_{n}$.*

## 7. Asymptotics

Strong and weak learning (or strong and weak knowledge acquisition) are equivalent for n=1. The larger *n* is, the more restrictive strong learning becomes in comparison to weak learning. However, for large *n*, neither strong nor weak learning (knowledge acquisition) are entirely satisfactory entities. For this reason, in this section we will introduce asymptotic versions of learning and knowledge acquisition, for an agent whose discernment between worlds and collected data sets increase over a long period of time.

### 7.1. Asymptotic Learning and Knowledge Acquisition

In order to define asymptotic learning and knowledge acquisition, as the number of steps *n* of the discernment and data collection process tends to infinity, we first need to introduce AIN versions of limits. Define
(40)$$\begin{array}{cc} I_{lim inf}^{+}\left(B\right) & =\log\frac{lim infP_{k}\left(B\right)}{P_{0}\left(B\right)}, \end{array}$$
(41)$$\begin{array}{cc} I_{lim sup}^{+}\left(B\right) & =\log\frac{lim supP_{k}\left(B\right)}{P_{0}\left(B\right)}, \end{array}$$
and when the two limits of (Equation 40) agree, we refer to the common value as $I_{\lim}^{+}\left(B\right)$. Define also
(42)$$\begin{array}{cc} {\bar{I}}_{lim inf}^{+}\left(B\right) & =\log\frac{lim inf{\bar{P}}_{k}\left(B\right)}{P_{0}\left(B\right)}, \end{array}$$
(43)$$\begin{array}{cc} {\bar{I}}_{lim sup}^{+}\left(B\right) & =\log\frac{lim sup{\bar{P}}_{k}\left(B\right)}{P_{0}\left(B\right)}, \end{array}$$
with ${\bar{I}}_{\lim}^{+}\left(B\right)$ the common value whenever the two limits of (Equation 42) agree. Since $I_{\lim}^{+}\left(B\right)$ only exists when $I_{lim inf}^{+}\left(B\right)=I_{lim sup}^{+}\left(B\right)$, and $I_{lim inf}^{+}\left(B\right)\leq I_{lim sup}^{+}\left(B\right)$, the following definitions of asymptotic learning and knowledge acquisition are natural:

**Definition** **7**(Asymptotic learning). *Learning occurs asymptotically (conditionally on the observed data sequence ${\left\{D_{k}\right\}}_{k=1}^{\infty }$) if*
(44)$$\begin{array}{c} \left\{\begin{array}{c} I_{\lim\inf}^{+}\left(A\right)>0,\;\mathrm{for}\;x_{0}\in A,\\ I_{\lim\sup}^{+}\left(A\right)<0,\;\;\mathrm{for}\;x_{0}\notin A. \end{array}\right. \end{array}$$*Full learning occurs asymptotically (conditionally on ${\left\{D_{k}\right\}}_{k=1}^{\infty }\}$) if*
(45)$$\begin{array}{c} \left\{\begin{array}{c} I_{\lim}^{+}\left(A\right)=-\log P_{0}\left(A\right),\;\mathrm{for}\;x_{0}\in A,\\ I_{\lim}^{+}\left(A\right)=-\infty ,\;\;\mathrm{for}\;x_{0}\notin A. \end{array}\right. \end{array}$$*Learning is expected to occur asymptotically if *(Equation 44)* holds with ${\bar{I}}_{lim sup}^{+}$ and ${\bar{I}}_{lim inf}^{+}$, instead of $I_{lim sup}^{+}$ and $I_{lim inf}^{+}$, respectively. Full learning is expected to occur asymptotically, if *(Equation 45)* holds with ${\bar{I}}_{\lim}^{+}$ instead of $I_{\lim}^{+}$.*

**Definition** **8**(Asymptotic knowledge acquisition). *Knowledge acquisition occurs asymptotically (conditionally on ${\left\{D_{k}\right\}}_{k=1}^{\infty }$) if, in addition to the asymptotic learning condition *(Equation 44)*, in order for this asymptotic learning to be justified, for every ϵ>0, it holds that*
$$\underset{k\to \infty }{lim inf}P_{k}\left(B_{\epsilon }\left(x_{0}\right)\right)>0$$*and*
(46)$$I_{lim inf}^{+}\left(B_{\epsilon }\left(x_{0}\right)\right)\geq 0,$$*with strict inequality for a least one ϵ>0. Full knowledge acquisition occurs asymptotically (conditionally on ${\left\{D_{k}\right\}}_{k=1}^{\infty }\}$) if *(Equation 45)* holds and*
(47)$$I_{\lim}^{+}\left(B_{\epsilon }\left(x_{0}\right)\right)=-\log P_{0}\left(B_{\epsilon }\left(x_{0}\right)\right)$$*is satisfied for all ϵ>0. If learning is expected to occur asymptotically according to Definition 7, and if *(Equation 46)* holds with ${\bar{I}}_{lim inf}^{+}$ instead of $I_{lim inf}^{+}$, then knowledge acquisition is expected to occur asymptotically. Full knowledge acquisition is expected to occur asymptotically if full learning is expected to occur asymptotically according to Definition 7, and if *(Equation 47)* holds with ${\bar{I}}_{\lim}^{+}$ instead of $I_{\lim}^{+}$.*

### 7.2. Bayesian Asymptotic Theory

In this subsection we will use Bayesian asymptotic theory in order to quantify and give conditions for when asymptotic learning and knowledge acquisition occur. Let Ω be a large space that incorporates prior beliefs and data for all k=1,2,…. Define $X_{k}:\Omega \to X$ as a random variable whose distribution corresponds to the agent’s posterior beliefs, based on data set $D_{k}$, which itself varies according to another random variable $D_{k}:\Omega \to D_{k}$ with distribution $D_{k}∼L\left(\cdot |x_{0}\right)$. Let ${\mathrm{Pr}}_{x_{0}}$ be a probability measure on subsets of Ω that induces distributions $X_{k}|D_{k}∼P_{k}$ and $X_{k}∼{\bar{P}}_{k}$, respectively. The following proposition is a consequence of Definitions 7 and 8:

**Proposition** **1.**
*Suppose full learning is expected to occur asymptotically, in the sense of *(Equation 45)*, with ${\bar{I}}_{\lim}^{+}$ instead of $I_{\lim}^{+}$. Then,*

(48)
$${Pr}_{x_{0}}\left(X_{k}\in A\right)={\bar{P}}_{k}\left(A\right)\to \left\{\begin{array}{cc} 1, & x_{0}\in A,\\ 0, & x_{0}\notin A \end{array}\right.$$

*as k→∞. In particular, the type I and II errors of the hypothesis test *(Equation 18)* and *(Equation 19)*, with threshold $I=\log[p/P_{0}\left(A\right)]$ for some 0<p<1, satisfy*

(49)
$$\begin{array}{ccc} \alpha_{k}\left(x_{0}\right) & = & {Pr}_{x_{0}}\left(I_{k}^{+}\left(A\right)\leq I\right)={Pr}_{x_{0}}\left[Pr\left(X_{k}\in A|D_{k}\right)\leq p\right]\\ & = & {Pr}_{x_{0}}\left(P_{k}\left(A\right)\leq p\right)\to 0,\;\;\;\;x_{0}\in A,\\ \beta_{k}\left(x_{0}\right) & = & {Pr}_{x_{0}}\left(I_{k}^{+}\left(A\right)>I\right)={Pr}_{x_{0}}\left[Pr\left(X_{k}\in A|D_{k}\right)>p\right]\\ & = & {Pr}_{x_{0}}\left(P_{k}\left(A\right)>p\right)\to 0,\;\;\;\;x_{0}\notin A, \end{array}$$

*respectively, as k→∞. If full knowledge acquisition occurs asymptotically, in the sense of *(Equation 47)*, then*

(50)
$$X_{k}|D_{k}\;\overset{p}{⟶}x_{0}conditionallyon{\left\{D_{k}\right\}}_{k=1}^{\infty }$$

*as k→∞, with $\;\overset{p}{⟶}$ referring to convergence in probability. If full knowledge acquisition is expected to occur asymptotically, in the sense of Definition 8, then*

(51)
$$X_{k}\;\overset{p}{⟶}x_{0}$$

*as k→∞.*


**Remark** **10.**
*Full asymptotic knowledge acquisition *(Equation 50)* is closely related to the notion of posterior consistency [40]. For our model, the latter concept is usually defined as*

(52)
$${Pr}_{x_{0}}\left(X_{k}|D_{k}\;\overset{p}{⟶}x_{0}ask\to \infty \right)=1,$$

*where the probability refers to variations in the data sequence ${\left\{D_{k}\right\}}_{k=1}^{\infty }$ when $x_{0}$ holds. Thus, posterior consistency *(Equation 52)* means that full asymptotic knowledge acquisition *(Equation 50)* holds with probability 1. Let L(X) refer to the distribution of the random variable X. Then, *(Equation 52)* is equivalent to*

(53)
$$P_{k}=L\left(X_{k}|D_{k}\right)\;\overset{a.s.}{⟶}\delta_{x_{0}}$$

*as k→∞, with $\;\overset{a.s.}{⟶}$ referring to almost sure weak convergence with respect to variations in the data sequence ${\left\{D_{k}\right\}}_{k=1}^{\infty }$ when $x=x_{0}$. On the other hand, it follows from Definition 8 that if full knowledge acquisition is expected to occur asymptotically, this is equivalent to*

(54)
$$P_{k}=L\left(X_{k}|D_{k}\right)\;\overset{p}{⟶}\delta_{x_{0}}$$

*as k→∞, which is a weaker concept than posterior consistency, since almost sure weak convergence implies weak convergence in probability. However, sometimes *(Equation 54)*, rather than *(Equation 52)* and *(Equation 53)*, is used as a definition of posterior consistency.*


**Remark** **11.**
*It is sometimes possible to sharpen *(Equation 54)* and obtain the rate at which the posterior distribution converges to $\delta_{x_{0}}$. The posterior distribution is said to contract at rate $\epsilon_{k}\to 0$ to $\delta_{x_{0}}$ as k→∞ (see for instance [41]), if for every sequence $M_{k}\to \infty$ it holds that*

(55)
$$Q\left[X_{k}|D_{k}\notin B\left(x_{0},M_{k}\epsilon_{k}\right)\right]=P_{k}\left[B{\left(x_{0},M_{k}\epsilon_{k}\right)}^{c}\right]\;\overset{p}{⟶}0,$$

*when ${\left\{D_{k}\right\}}_{k=1}^{\infty }$ varies according to what one expects in the true world $x_{0}$. Since convergence in probability is equivalent to convergence in mean for bounded random variables, it can be seen that *(Equation 54)* is equivalent to ${\bar{P}}_{k}\left[B{\left(x_{0},M_{k}\epsilon_{k}\right)}^{c}\right]\to 0$, or*

(56)
$${\left(M_{k}\epsilon_{k}\right)}^{-1}\left(X_{k}-x_{0}\right)\;\overset{p}{⟶}0$$

*as k→∞ for each sequence $M_{k}\to \infty$. Comparing *(Equation 51)* with *(Equation 55)* and *(Equation 56)*, we found that a contraction of the posterior towards $\delta_{x_{0}}$ at rate $\epsilon_{k}$ is equivalent to expecting full knowledge acquisition asymptotically at rate $\epsilon_{k}$.*


It follows from Proposition 1 and Remarks 10 and 11 that Bayesian asymptotic theory can be used, within our frequentist/Bayesian framework, to give sufficient conditions for asymptotic learning and knowledge acquisition to occur. Suppose, for instance, that $D_{k}=\left(Z_{1},\ldots ,Z_{k}\right)$ is a sample of *k* independent and identically distributed random variables $Z_{l}$ with distribution $Z_{l}∼F\left(\cdot |x_{0}\right)$ that belongs to the statistical model {F(·∣x);x∈X}. The likelihood function is then a product
(57)$$L\left(D_{k}|x\right)=\prod_{l=1}^{k}Pr\left(Z_{l}|x\right)$$
of the likelihoods of all observations $Z_{l}$. For such a model, a number of authors [40,42,43,44,45] have provided sufficient conditions for posterior consistency (Equation 52) and (Equation 53) to occur. It follows from Remark 10 that these conditions also imply the weaker concept (Equation 54) of full, expected knowledge acquisition to occur asymptotically.

Suppose (Equation 57) holds with a parameter space $X\subset R^{q}$ that is a subset of Euclidean space of dimension *q*. It is possible then to obtain the rate (Equation 56) at which knowledge acquisition is expected to occur. The first step is to use the Bernstein–von Mises theorem, which under appropriate conditions (see for instance [46]) approximates the posterior distribution $P_{k}=L\left(X_{k}|D_{k}\right)$ by a normal distribution centered around the maximum likelihood (ML) estimator
(58)$${\hat{x}}_{0k}={\hat{x}}_{0k}\left(D_{k}\right)={argmax}_{x\in X}L\left(D_{k}|x\right)$$
of $x_{0}$. More specifically, this theorem provides weak convergence
(59)$$\sqrt{k}\left(X_{k}-{\hat{x}}_{0k}\right)|D_{k}\;\overset{L}{⟶}N\left(0,J{\left(x_{0}\right)}^{-1}\right)$$
as k→∞, of a re-scaled version of the distribution of $X_{k}|D_{k}$ when ${\left\{D_{k}\right\}}_{k=1}^{\infty }$ varies according what one expects when $x=x_{0}$. The limiting distribution is a *q*-dimensional normal distribution with mean 0 and a covariance matrix that equals the inverse of the Fisher information matrix $J(x_{0})$, evaluated at the true world parameter $x_{0}$. On the other hand, the standard asymptotic theory of maximum likelihood estimation (see for instance [47]) implies
(60)$$\sqrt{k}\left({\hat{x}}_{0k}-x_{0}\right)\;\overset{L}{⟶}N\left(0,J{\left(x_{0}\right)}^{-1}\right)$$
as k→∞, with weak convergence referring to variations in the data sequence ${\left\{D_{k}\right\}}_{k=1}^{\infty }$ when $x=x_{0}$. Combining equations (Equation 59) and (Equation 60), we arrive at the following result:

**Theorem** **1.**
*Assume data ${\left\{D_{k}=\left(Z_{1},\ldots ,Z_{k}\right)\right\}}_{k=1}^{\infty }$ consists of independent and identically distributed random variables ${\left\{Z_{l}\right\}}_{l=1}^{\infty }$, and that the Bernstein–von Mises theorem *(Equation 59)* and asymptotic normality *(Equation 60)* of the ML-estimator hold. Then, $X_{k}$ converges weakly towards $x_{0}$ at rate $1/\sqrt{k}$, in the sense that*

(61)
$$\sqrt{k}\left(X_{k}-x_{0}\right)\;\overset{L}{⟶}N\left(0,2J{\left(x_{0}\right)}^{-1}\right)$$

*as k→∞. In particular, full knowledge acquisition is expected to occur asymptotically at rate $1/\sqrt{k}$.*


**Proof.** Let $s\to F_{k}\left(s\right)=F_{\sqrt{k}\left({\hat{x}}_{k0}-x_{0}\right)}\left(s\right)$ refer to the distribution function of $\sqrt{k}\left({\hat{x}}_{k0}\left(D_{k}\right)-x_{0}\right)$, defined for all *q*-dimensional vectors $s=\left(s_{1},\ldots ,s_{q}\right)\in R^{q}$. Let also $t=\left(t_{1},\ldots ,t_{q}\right)\in R^{q}$ and denote the distribution function of $N\left(0,2J{\left(x_{0}\right)}^{-1}\right)$ by *G*. Combining (Equation 59) and (Equation 60), and making use of the fact that the convolution of two independent $N\left(0,J{\left(x_{0}\right)}^{-1}\right)$-variables is distributed as $N\left(0,2J{\left(x_{0}\right)}^{-1}\right)$, we can find that
(62)$$\begin{array}{ccc} {Pr}_{x_{0}}\left(\sqrt{k}\left(X_{k}-x_{0}\right)\leq t\right) & = & \int {Pr}_{x_{0}}\left(\sqrt{k}\left(X_{k}-{\hat{x}}_{0k}\right)\leq t-s|\sqrt{k}\left({\hat{x}}_{0k}-x_{0}\right)=s\right)dF_{k}\left(s\right)\\ \; & \to & \int_{s\leq t}dG\left(s\right) \end{array}$$
as k→∞, with s≤t referring to $s_{j}\leq t_{j}$ for j=1,…,q. Since (Equation 62) holds for any $t\in R^{q}$, Equation (Equation 61) follows. Moreover, in view of (Equation 56), Equation (Equation 61) implies that full knowledge acquisition is expected to occur asymptotically at rate $1/\sqrt{k}$. □

In general, the conditions of Theorem 1 typically require that data, and the agent’s interpretation of data, are unbiased. When these conditions fail (cf. Remark 2), there is no guarantee that knowledge acquisition is expected to occur asymptotically as k→∞.

## 8. Examples

**Example** **1**(Coin tossing). *Let $x_{0}\in X=\left[0,1\right]$ be the probability of heads when a certain coin is tossed. An agent wants to find out whether the proposition*
S:the coin is symmetric with margin ε>0*is true or not. This corresponds to a truth function f(x)=1(x∈A), with A=[0.5−ε,0.5+ε], that is known to the agent. Suppose the coin is tossed a large number of times (n=∞), and let $D_{k}=\left(Z_{1},\ldots ,Z_{k}\right)\in D_{k}={\left\{0,1\right\}}^{k}$ be a binary sequence of length k that represents the first k tosses, with tails and heads corresponding to 0 ($Z_{k}=0$) and 1 ($Z_{k}=1$), respectively. The number of heads $M_{k}=\sum_{l=1}^{k}Z_{l}∼Bin\left(k,x_{0}\right)$ after k tosses is then a sufficient statistic for estimating $x_{0}$ based on data $D_{k}$. Even though $\left\{D_{k}\right\}$ is an increasing sequence of data sets, we put $Y_{k}\left(x\right)=x$ and $G_{k}=F=B\left(\left[0,1\right]\right)$, the Borel σ-algebra on [0,1], for k=1,2,…. Let $P_{0}$ be the uniform prior distribution on [0,1]. Since the uniform distribution is a beta distribution, and beta distributions are conjugate priors to binomial distributions, it is well known [17] that the posterior distribution*
$$P_{k}∼\mathrm{Beta}\left(1+M_{k},1+k-M_{k}\right)$$*belongs to the beta family as well. Consequently, if $X_{k}$ is a random variable that reflects the agent’s degree of belief in the probability of heads after k tosses, it follows that his belief in a symmetric coin, if $M_{k}=m$, is*
$$\begin{array}{cc} P_{k}\left(A\right)=Pr\left(X_{k}\in A|D_{k}\right) & =Pr\left(X_{k}\in A|M_{k}=m\right)\\ \; & =\int_{0.5-\varepsilon }^{0.5+\varepsilon }p_{k}\left(x|m\right)dx, \end{array}$$*where*
(63)pkx∣m=(1−x)k−mxmB(1+m,1+k−m)=(k+1)km(1−x)k−mxm=(k+1)Lm∣x*is the posterior density function of the parameter x, whereas B(a,b) is the beta function and x→L(m∣x) the likelihood function. From this, it follows that the AIN after k coin tosses with m heads and k−m tails equals*
Ik+(A)=log[(2ε)−1Pk(A)]=log(2ε)−1(k+1)km∫0.5−ε0.5+ε(1−x)k−mxmdx.
*Since data are random, $P_{k}\left(A\right)$ (and hence also $I_{k}^{+}\left(A\right)$) will fluctuate randomly up and down with probability one (see Figure 3); for this reason, ${\left\{P_{k}\right\}}_{k=1}^{\infty }$ does not represent a learning process in the strong sense of Definition 3. On the other hand, it follows by the strong law of large numbers that $M_{k}/k\overset{a.s.}{⟶}x_{0}$ as k→∞, and from properties of the beta distribution, this implies that full learning and knowledge acquisition occur asymptotically according to Definitions 7 and 8, with probability 1. In view of Remark 10, we also have posterior consistency *(Equation 52)* and *(Equation 53)*.*

*By analyzing ${\bar{P}}_{k}$ instead of $P_{k}$, we may also assess whether learning and knowledge acquisition are expected to occur. The expected degree of belief in a symmetric coin, after k tosses, is*

$${\bar{P}}_{k}\left(A\right)=\int_{0.5-\varepsilon }^{0.5+\varepsilon }{\bar{p}}_{k}\left(x\right)dx,$$

*where*

$$\begin{array}{cc} {\bar{p}}_{k}\left(x\right) & =E_{x_{0}}\left[p_{k}\left(x|M_{k}\right)\right]\\ \; & =\sum_{m=0}^{k}L\left(m|x_{0}\right)p_{k}\left(x|m\right)\\ \; & =\left(k+1\right)\sum_{m=0}^{k}L\left(m|x_{0}\right)L\left(m|x\right) \end{array}$$

*is the expected posterior density function of x, after k tosses of the coin. Note in particular that*

$$\int_{0}^{1}{\bar{p}}_{k}\left(x\right)dx=1.$$


*It can be shown that *(Equation 63)* and the weak law of large numbers ($M_{k}/k\overset{p}{⟶}x_{0}$ as k→∞, where $\overset{p}{⟶}$ refers to convergence in probability) lead to uniform convergence*

$$\underset{x:|x-x_{0}|\geq \epsilon }{\sup}{\bar{p}}_{k}\left(x\right)\to 0$$

*as k→∞ for any ϵ>0. The last four displayed equations imply ${\bar{P}}_{k}\left(A\right)\to 1\left(x_{0}\in A\right)$ and ${\bar{P}}_{k}\overset{p}{⟶}x_{0}$ as k→∞. This and Definitions 7 and 8 imply that full learning and knowledge acquisition are expected to occur asymptotically. This result is also a consequence of posterior consistency, or of Theorem 1. Notice, however, that a purely Bayesian analysis does not allow us to conclude that knowledge acquisition occurs, or is expected to occur, asymptotically.*


**Example** **2**(Historical events).
*Let X=(0,1] represent a time interval of the past. A person wants to find out whether his ancestor died or not during a famine that occurred in the province where the ancestor lived. Formally, this corresponds to a proposition*
S:the ancestor died during the time of the famine.
*Let f(x)=1 if the famine occurred at time x, and f(x)=0 if not. Assume that the ancestor died at an unknown time point $x_{0}$ and that the time period during which the famine lasted is A=[a,b], where 0≤a<b≤1 are known. If X corresponds to a fairly short time interval of the past, it is reasonable to assume that $P_{0}$ has a uniform distribution on (0,1].*

*In the first step of the learning process, suppose radiometric dating $D_{1}=Z_{1}$ of a burial find from the ancestor appears. If δ=1/N represents the precision of this dating method, the corresponding σ-algebra is*

(64)
$$\begin{array}{cc} G_{1} & =\sigma ((0,1/\delta ],(1/\delta ,2/\delta ],\ldots ,[(N-1)/\delta ,1])\\ \; & =\sigma (Y_{1}), \end{array}$$

*where $Y_{1}:X\to \left\{1,\ldots ,N\right\}$ is defined through $Y_{1}\left(x\right)=\left[x/\delta \right]+1$, and where [x/δ] is the integer part of x/δ. Due to *(Equation 3)*, it follows that $P_{1}$ has a density function*

(65)
$$p_{1}\left(x\right)=N\sum_{i=1}^{N}p_{1i}1\left(x\in \left(\left(i-1\right)\delta ,i\delta \right]\right),$$

*for some non-negative probability weights $p_{1i}\geq 0$ that sum to 1. Since $p_{1i}=p_{1i}\left(D_{1}\right)$, this measure is constructed from the radiometric dating data $D_{1}$ of the burial find from the ancestor. The $G_{1}$-optimal probability measure is obtained from *(Equation 8)* as*

$$p_{1i}=\left\{\begin{array}{cc} 1, & i=i_{0}=\left[x_{0}/\delta \right]+1,\\ 0, & i\neq i_{0}=\left[x_{0}/\delta \right]+1. \end{array}\right.$$

*It corresponds to dating the time of death of the ancestor correctly, given the accuracy of this dating method. On the other hand, if the radiometric dating equipment has a systematic error of −δ, a truth-excluding probability measure *(Equation 10)* is obtained with*

(66)
$$p_{1i}=\left\{\begin{array}{cc} 1, & i=i_{0}-1=\left[x_{0}/\delta \right],\\ 0, & i\neq i_{0}-1=\left[x_{0}/\delta \right]. \end{array}\right.$$


*In the second step of the learning process, suppose data $D_{2}=\left(Z_{1},Z_{2}\right)$ is extended to include a piece of text $Z_{2}$ from a book where the time of death of the ancestor can be found. This extra source of information increases the σ-algebra to $G_{2}=F=B\left(\left[0,1\right]\right)$, and if the contents of the book are reliable, $P_{2}=\delta_{x_{0}}$ is F-optimal. It follows from Definition 3 that strong learning has occurred if $Na=i_{a}$ and $Nb=i_{b}$ are integers and*

(67)
$$\begin{array}{cc} 0<I_{1}^{+}=\log\left[\sum_{i=i_{a}+1}^{i_{b}}p_{1i}/\left(b-a\right)\right]<I_{2}^{+}=\log\left[1/\left(b-a\right)\right], & ifx_{0}\in \left(a,b\right),\\ 0>I_{1}^{+}=\log\left[\sum_{i=i_{a}+1}^{i_{b}}p_{1i}/\left(b-a\right)\right]>I_{2}^{+}=-\infty , & ifx_{0}\notin \left(a,b\right). \end{array}$$

*Figure 4 illustrates another scenario where not only strong learning but also strong knowledge acquisition occurs. Suppose now that *(Equation 66)* holds, with $i_{a}+1\leq i_{0}-1\leq i_{b}$. If $P_{2}=\delta_{x_{0}}$, the strong learning condition *(Equation 67)* is satisfied, and the weak knowledge acquisition requirement of Definition 6 holds as well. Strong knowledge acquisition has not occurred though, since $p_{1i_{0}}=0$ means that Equation *(Equation 34)* of Definition 4 (with n=2) is violated for sufficiently small ϵ>0. Note in particular that these conclusions about knowledge acquisition cannot be drawn from a purely Bayesian analysis.*

*Assume now that the contents of the book are not reliable. A probability measure $P_{2}$ on [0,1] may be chosen so that it incorporates data $Z_{1}$ from the radiometric dating and data $Z_{2}$ from the book. This probability measure will also include information about the way the text of the book is believed to be unreliable. If the agent trusts $Z_{2}$ too much, it may happen that strong learning does not occur.*


**Example** **3**(Future events). *A youth camp with certain outdoor activities is planned for a weekend. Let $X={\left(0,1\right]}^{2}$ denote the set of possible temperatures $x=(x_{1},x_{2})$ of the two days for which the camp is planned, each normalized within a range $0\leq x_{i}\leq 1$. The outdoor activities are only possible within a certain sub-range $0<a\leq x_{1},x_{2}\leq b<1$ of temperatures. The proposition*
S:itispossibletohavetheoutdooractivities*corresponds to a truth function $f\left(x\right)=1\left(x\in {\left[a,b\right]}^{2}\right)$ and $A={\left[a,b\right]}^{2}$. The leaders have to travel to the camp five days before it starts and then make a decision on whether to bring equipment for the outdoor activities or for some other indoor activities. In the first step they consult weather forecast data $D_{1}=Z_{1}$, with a σ-algebra $G_{1}$ given by*
$$\sigma \left\{\left(\left(i-1\right)\delta_{1},i\delta_{1}\right]\times \left(\left(j-1\right)\delta_{2},j\delta_{2}\right];\;1\leq i\leq N_{1},1\leq j\leq N_{2}\right\},$$*which is $\sigma (Y_{1})$, the σ-algebra generated by $Y_{1}$, where $\delta_{1}$ and $\delta_{2}>\delta_{1}$ represent the maximal possible accuracy of weather forecasts five and six days ahead, respectively, $N_{i}=1/\delta_{i}$ and $Y_{1}\left(x\right)=\left(\left[x_{1}/\delta_{1}\right]+1,\left[x_{2}/\delta_{2}\right]+1\right)$. Let $P_{0}$ be the uniform distribution on ${\left[0,1\right]}^{2}$. Due to *(Equation 3)*, $P_{1}$ has a density function*
(68)$$p_{1}\left(x\right)=N_{1}N_{2}\sum_{i=1}^{N_{1}}\sum_{j=1}^{N_{2}}p_{1ij}1\left(\left(x_{1},x_{2}\right)\in R_{ij}\right),$$*for some non-negative probability weights $p_{1ij}=p_{1i}\left(D_{1}\right)\geq 0$ that sum to 1, with*
$$\begin{array}{c} R_{ij}=\left(\left(i-1\right)\delta_{1},i\delta_{1}\right]\times \left(\left(j-1\right)\delta_{2},j\delta_{2}\right], \end{array}$$*a rectangular region that corresponds to the ith temperature interval the first day of the camp and the jth temperature interval the second day. Consequently, the accuracy $G_{1}$ of weather forecast data forces $p_{1}$ to be constant over each $R_{ij}$. A $G_{1}$-optimal measure assigns full weight 1 to the rectangle $R_{ij}$ with $i=[x_{01}/\delta_{1}]+1$ and $j=[x_{02}/\delta_{2}]+1$, where $x_{0}=\left(x_{01},x_{02}\right)$ represents the actual temperature the two days. Observe then that the $G_{1}$-optimal measure is restricted to measurements that are accurate up to $\delta_{1}$ and $\delta_{2}$; therefore, it cannot do better than assigning the temperature to the intervals with sizes $\delta_{1}$ and $\delta_{2}$ to which the actual temperatures belong; however, it cannot say what the exact temperature is. The exact prediction requires an F-optimal measure.*
*In a second step, in order to get some additional information, the leaders of the camp consult a prophet. Let $P_{2}$ refer to the probability measure based on the weather forecast $Z_{1}$ and the message $Z_{2}$ of the prophet, so that $D_{2}=\left(Z_{1},Z_{2}\right)$ and $G_{2}=F$. If the prophet always speaks the truth, and if the leaders of the camp rely on his message, they will make use of the F-optimal measure $P_{2}=\delta_{x_{0}}$, corresponding weak (and full) learning, and a full amount of knowledge. In general, the camp leaders’ prediction in step k is correct with*

$$probability=\left\{\begin{array}{cc} P_{k}\left({\left[a,b\right]}^{2}\right), & if\;x_{0}\in {\left[a,b\right]}^{2},\\ 1-P_{k}\left({\left[a,b\right]}^{2}\right), & if\;x_{0}\notin {\left[a,b\right]}^{2}. \end{array}\right.$$

*If this probability is less than 1 for k=2, the reason is either that the prophet does not always speak the truth or that the leaders do not rely solely on the message of the prophet. In particular, it follows from Definition 3 that strong learning has occurred if*

(69)
$$\begin{array}{cc} 0<I_{1}^{+}=\log\left[P_{1}\left({\left[a,b\right]}^{2}\right)/{\left(b-a\right)}^{2}\right]<I_{2}^{+}=\log\left[P_{2}\left({\left[a,b\right]}^{2}\right)/{\left(b-a\right)}^{2}\right], & ifx_{0}\in {\left[a,b\right]}^{2},\\ 0>I_{1}^{+}=\log\left[P_{1}\left({\left[a,b\right]}^{2}\right)/{\left(b-a\right)}^{2}\right]>I_{2}^{+}=\log\left[P_{2}\left({\left[a,b\right]}^{2}\right)/{\left(b-a\right)}^{2}\right], & ifx_{0}\notin {\left[a,b\right]}^{2}. \end{array}$$

*Suppose the weather forecast and the message of the prophet are biased, but they still correctly predict whether outdoor activities are possible or not. Then, neither weak nor strong knowledge acquisition occurs, in spite of the fact that the strong learning condition *(Equation 69)* holds. Note in particular that such a conclusion is not possible with a purely Bayesian analysis. Another scenario wherein neither (weak or strong) learning nor knowledge acquisition takes place is depicted in Figure 5.*


**Example** **4**(Replication of studies). *Some researchers want to find the prevalence of the physical symptoms of a certain disease. Let $X={\left[0,1\right]}^{2}$ refer to the possible set of values $x=(x_{1},x_{2})$ for the prevalence of the symptoms, obtained from two different laboratories. The first value corresponds to the prevalence obtained in Laboratory 1, whereas the second value $x_{2}$ is obtained when Laboratory 2 tries to replicate the study of Laboratory 1. The board members of the company to which the two laboratories belong want to find out whether the two estimates are consistent, within some tolerance level 0<ε<1. In that case, the second study is regarded as replication of the first one. The proposition*
S:the second study replicates the first one*corresponds to a truth function $f\left(x\right)=1\left(|x_{2}-x_{1}|\leq \varepsilon \right)$ and*
(70)$$A=\left\{\left(x_{1},x_{2}\right);\;\left|x_{2}-x_{1}\right|\leq \varepsilon \right\}.$$
*The true value $x_{0}=\left(x_{01},x_{02}\right)$ represents the actual prevalences of the symptoms, obtained from the two laboratories under ideal conditions. Importantly, it may still be the case that $x_{01}\neq x_{02}$, if either the prevalence of the symptoms changes between the two studies and/or the two laboratories estimate the prevalences within two different subpopulations.*

*Let $D_{2}$ be a data set by which Laboratory 2 receives all needed data from Laboratory 1 in order to set up its study properly (so that, for instance, affection status is defined in the same way in the two laboratories ). We will assume $Y_{2}\left(x_{1},x_{2}\right)=\left(x_{1},x_{2}\right)$, so that the corresponding σ-algebra*

$$G_{2}=F=B\left(X\right)=B_{0}\times B_{0},$$

*corresponds to full discernment, with B(X) being the Borel σ-algebra on X, whereas $B_{0}=B\left(\left(0,1\right]\right)$ is the Borel σ-algebra on the unit interval (see Remark 1). If $P_{2}$ is the probability measure obtained from $D_{2}$, the probability of concluding that the second study replicated the first is*

(71)
$$P_{2}\left(A\right)=\int_{A}dP_{2}\left(x_{1},x_{2}\right).$$


*In particular, when each laboratory makes use of data from all individuals in its subpopulation (which is either the same or not for the two laboratories), the F-optimal probability measure (Equation 9) corresponds to*

(72)
$$P_{2}=\delta_{x_{0}}⟹P_{2}\left(A\right)=1(|x_{02}-x_{01}\left|\leq \varepsilon \right).$$


*Now consider another scenario where Laboratory 2 only gets partial information from Laboratory 1. This corresponds to a data set $D_{1}$ with the same sampled individuals as in $D_{2}$, but Laboratory 2 has incomplete information from Laboratory 1 regarding the details of how the first study was set up. For this reason, they make use of a coarser σ-algebra, by which it is only possible to quantify prevalence with precision δ. If this σ-algebra is referred to as $B_{\delta }\subset B_{0}$, it follows that $Y_{1}\left(x_{1},x_{2}\right)=\left(x_{1},\left[x_{2}/\delta \right]+1\right)$ and*

$$G_{1}=B_{0}\times B_{\delta }.$$


*The corresponding loss of information is measured through a probability $P_{1}$ that has the same marginal distribution as $P_{2}$ for all events B that are discernible from $G_{1}$, i.e.,*

(73)
$$P_{1}\left(B\right)=P_{2}\left(B\right),\;\forall B\in G_{1}.$$

*Hence, it follows from *(Equation 30)* and *(Equation 73)* that*

$$dP_{1}\left(x_{1},x_{2}\right)=N\sum_{j=1}^{N}p_{1j}\left(x_{1}\right)1\left(x_{2}\in R_{j}\right)dP_{2}\left(x_{1}\right),$$

*where N=1/δ, $p_{1j}\left(x_{1}\right)=P_{2}\left(X_{2}\in R_{j}|X_{1}=x_{1}\right)$, and $R_{j}=\left(\left(j-1\right)\delta ,j\delta \right]$ is the j-th possible region for the prevalence estimate of Laboratory 2. In particular, the probability that the second study replicates the first one is*

(74)
$$P_{1}\left(A\right)=N\int_{0}^{1}\sum_{j=1}^{N}p_{1j}\left(x_{1}\right)\left|R_{j}\cap \left[x_{1}-\varepsilon ,x_{1}+\varepsilon \right]\right|dP_{2}\left(x_{1}\right).$$


*If both laboratories perform a screening and collect data from all individuals in their regions, so that *(Equation 72)* holds, then $P_{1}$ is a $G_{1}$-optimal measure according to *(Equation 8)*, with*

(75)
$$P_{1}\left(A\right)=N\left|R_{j_{0}}\cap \left[x_{01}-\varepsilon ,x_{01}+\varepsilon \right]\right|,$$

*and $j_{0}=\left[x_{02}/\delta \right]+1$. Making use of Definition 3, we notice that a sufficient condition for strong learning to occur is that $P_{0}$ has a uniform distribution on X (so that $P_{0}\left({\left[a,b\right]}^{2}\right)=2\varepsilon -\varepsilon^{2}$), such that *(Equation 72)* and *(Equation 75)* hold, and*

$$\begin{array}{cc} 0<I_{1}^{+}=\log[N|R_{j_{0}}\cap \left[x_{01}-\varepsilon ,x_{01}+\varepsilon \right]\left|/\left(2\varepsilon -\varepsilon^{2}\right)\right]<I_{2}^{+}=1, & if|x_{02}-x_{01}|<\varepsilon ,\\ 0>I_{1}^{+}=\log[N|R_{j_{0}}\cap \left[x_{01}-\varepsilon ,x_{01}+\varepsilon \right]\left|/\left(2\varepsilon -\varepsilon^{2}\right)\right]>I_{2}^{+}=-\infty , & if|x_{02}-x_{01}|>\varepsilon . \end{array}$$

*With full information transfer between the two laboratories, the replication probabilities *(Equation 71)* and *(Equation 72)* based on data $D_{2}$ only depend on ε, whereas the corresponding replication probabilities *(Equation 74)* and *(Equation 75)* under incomplete information transfer between the laboratories and data $D_{1}$, also depending on δ. In particular, $P_{1}\left(A\right)$ will always be less than 1 when 2ε<δ, even when (Equation 75) holds and $x_{01}=x_{02}$. Moreover, δ sets the limit in terms of how much knowledge can be obtained from the two studies under incomplete information transfer, since*

$$P_{1}\left(B_{\epsilon }\left(x_{0}\right)\right)<1,\;\mathrm{for all}\;0<\epsilon <\delta .$$

*Note that this last conclusion cannot be obtained from a Bayesian analysis, since a true pair $x_{0}$ of prevalences does not belong to a purely Bayesian framework.*


**Example** **5**(Unmeasured confounding and causal inference). *This example illustrates unmeasured confounding and causal inference. Let q=n and $X={\left\{0,1\right\}}^{n}$. An individual is assigned a binary vector $x=(x_{1},\ldots ,x_{n})$ of length n, where $x_{n}\in \left\{0,1\right\}$ codes for whether that person will have symptoms within five years ($x_{n}=1$) or not ($x_{n}=0$) that are associated with a certain mental disorder. The first component $x_{1}\in \left\{0,1\right\}$ refers to the individual’s binary exposure, whereas the other variables $x_{k}\in \left\{0,1\right\}$, k=2,…,n−1 are binary confounders. The truth function $f\left(x\right)=x_{n}$ corresponds to symptom status, whereas*
$$A=\{x\in X;x_{n}=1\}$$*represents the vectors x of all individuals in the population with symptoms. Consider the proposition*
S:Adam will have the symptoms within five years,*and let $x_{0}=\left(x_{01},\ldots ,x_{0n}\right)$ be the vector associated with Adam. We will introduce a sequence of probability measures $P_{0},P_{1},\ldots ,P_{n}$, where $P_{0}$ represents the distribution of $X=(X_{1},\ldots ,X_{n})$ in the whole population, whereas $P_{k}$ corresponds to the conditional distribution of $X∼P_{0}$, given that its first k covariates $D_{k}=\left(Z_{1},\ldots ,Z_{k}\right)=\left(x_{01},\ldots ,x_{0k}\right)\in D_{k}={\left\{0,1\right\}}^{k}$ have been observed, with values equal to those of Adam’s first k covariates. Since the conditional distribution $D_{k}|x_{0}$ is non-random, it follows that*
(76)$${\bar{P}}_{k}=P_{k}=\prod_{l=1}^{k}\delta_{x_{0l}}\times P_{0}\left(\;\cdot |{\left\{X_{l}=x_{0l}\right\}}_{l=1}^{k}\right)$$*for k=0,1,…,n−1, whereas ${\bar{P}}_{n}=P_{n}=\delta_{x_{0}}$ for k=n. According to Definition 5, this implies that weak learning occurs with probability 1, and in particular that weak learning is expected to occur. If $Y_{k}\left(x_{1},\ldots ,x_{n}\right)=\left(x_{1},\ldots ,x_{k}\right)$, we have that*
(77)$$G_{k}=2^{{\left\{0,1\right\}}^{k}}\times {\left\{0,1\right\}}^{n-k}$$*for k=0,…,n. Note, in particular, that $P_{k}$ is $G_{k}$-optimal, corresponding to error-free measurement of Adam’s first k covariates.*
*In order to specify the null distribution $P_{0}$, we assume that a logistic regression model [48]*

(78)
$$\begin{array}{cc} P_{0}\left(X_{n}=1|x_{1},\ldots ,x_{n-1}\right) & =\frac{\exp\left(\beta_{0}+\sum_{k=1}^{n-1}\beta_{k}x_{k}\right)}{1+\exp\left(\beta_{0}+\sum_{k=1}^{n-1}\beta_{k}x_{k}\right)}\\ \; & =g(x_{1},\ldots ,x_{n-1}) \end{array}$$

*holds for the probability of having the symptoms within five years, conditionally on the n−1 covariates (one exposure and n−2 confounders). It is also assumed that the regression parameters $\beta_{0},\ldots ,\beta_{n-1}$ are known, so that g is known as well. It follows from Equations *(Equation 76)* and *(Equation 78)* that*

(79)
$$\begin{array}{c} \begin{array}{cc} P_{k}\left(A\right) & =P_{0}\left(X_{n}=1|{\left\{X_{l}=x_{0l}\right\}}_{l=1}^{k}\right)\\ \; & =E_{P_{0}}\left[g\left(X_{1},\ldots ,X_{n-1}\right)|{\left\{X_{l}=x_{0l}\right\}}_{l=1}^{k}\right]\\ \; & =:g_{k}\left(x_{01},\ldots ,x_{0k}\right) \end{array} \end{array}$$

*can be interpreted as increasingly better predictions of Adam’s symptom status five years ahead, for k=0,1,…,n−1, whereas $P_{n}\left(A\right)=f\left(x_{0}\right)=x_{0n}$ represents full knowledge of S. In particular, $P_{0}\left(A\right)$ is the prevalence of the symptoms in the whole population, whereas $P_{1}\left(A\right)=g_{1}\left(x_{01}\right)$ is Adam’s predicted probability of having the symptoms when his exposure $x_{01}$ is known, whereas none of his confounders are measured.*

*Suppose $x_{2},\ldots ,x_{n-1}$ are sufficient for confounding control, and that the exposure and the confounders (in principle) can be assigned. Let $x_{0}=\left(x_{01},\ldots ,x_{0n}\right)$ represent a hypothetical individual for which all covariates are assigned. Under a so called conditional exchangeability condition [16], it is possible to use a slightly different definition*

$${\tilde{P}}_{k}=\prod_{l=1}^{k}\delta_{x_{0l}}\times E_{P_{0}}\left[P_{0}\left(\cdot |{\left\{X_{l}\right\}}_{l=1}^{k}\right)\right]$$

*of the probability measures in order to compute the counterfactual probability*

$$\begin{array}{cc} h_{k}\left(x_{01},\ldots ,x_{0k}\right) & ={\tilde{P}}_{k}\left(A\right)\\ \; & =E_{P_{0}}\left[g\left(x_{01},\ldots ,x_{0k},X_{k+1},\ldots ,X_{n-1}\right)\right] \end{array}$$

*of the potential outcome $X_{n}=1$, under the scenario that the first k covariates were set to $x_{01},\ldots ,x_{0k}$. In particular, it is of interest to know how much the unknown causal risk ratio effect $h_{1}\left(1\right)/h_{1}\left(0\right)$ of the exposure maximally differs from the known risk ratio $g_{1}\left(1\right)/g_{0}\left(0\right)$ [49,50,51,52]. Note in particular that the corresponding logged quantities*

$$\begin{array}{ccc} \log[g_{1}\left(1\right)/g_{1}\left(0\right)] & = & I_{1}^{+}\left(A;1\right)-I_{1}^{+}\left(A;0\right),\\ \log[h_{1}\left(1\right)/h_{1}\left(0\right)] & = & {\tilde{I}}_{1}^{+}\left(A;1\right)-{\tilde{I}}_{1}^{+}\left(A;0\right), \end{array}$$

*can be expressed in terms of the active information*

$$\begin{array}{ccc} I_{1}^{+}\left(A;x_{01}\right) & = & \log[P_{1}\left(A\right)/P_{0}\left(A\right)]\\ & = & \log[P_{0}\left(X_{n}=1|x_{01}\right)/P_{0}\left(X_{n}=1\right)],\\ {\tilde{I}}_{1}^{+}\left(A;x_{01}\right) & = & \log[{\tilde{P}}_{1}\left(A\right)/P_{0}\left(A\right)]\\ & = & \log[E_{P_{0}}\left(P_{0}\left(X_{n}=1|x_{01},X_{2},\ldots ,X_{n-1}\right)\right)/P_{0}\left(X_{n}=1\right)]. \end{array}$$



## 9. Discussion

In this paper, we studied an agent’s learning and knowledge acquisition within a mathematical framework of possible worlds. Learning is interpreted as an increased degree of true belief, whereas knowledge acquisition additionally requires that the belief is justified, corresponding to an increased belief in the correct world. The theory is put into a framework that involves elements of frequentism and Bayesianism, with possible worlds corresponding to the parameters of a statistical model, where only one parameter value is true, whereas the agent’s beliefs are obtained from a posterior distribution. We formulated learning as a hypothesis test within this framework, whereas knowledge acquisition corresponds to consistency of posterior distributions. Importantly, we argue that a hybrid frequentist/Bayesian approach is needed in order to model mathematically the way in which philosophers distinguish learning from knowledge acquisition.

Some applications of our theory were provided in the examples of Section 8. Apart from those, we argue that our framework has quite general implications for machine learning, in particular, supervised learning. A typical task of machine learning is to obtain a predictor of a binary outcome variable $Y=f(x_{0})$, when only incomplete information *X* of $x_{0}$ is obtained from training data. The performance of a machine learning algorithm is typically assessed in terms of prediction accuracy, that is, how well f(X) approximates *Y*, with less focus on the closeness between *X* and $x_{0}$. In our terminology, the purpose of machine learning is learning rather than knowledge acquisition. This can often be a disadvantage, since knowledge acquisition often provides deeper insights than learning. For instance, full knowledge acquisition may fail asymptotically when k→∞, even when data are unbiased and interpreted correctly by the agent, if there is lacking discernment between the set of possible worlds X, even in the limit k→∞.

On the other hand, it makes no sense to go beyond learning for game theory, where the purpose is to find the optimal strategy (an instance of knowledge-how). In more detail, let x∈X={0,…,M−1} refer to the strategy *x* of a player among a finite set of *M* possible strategies. The optimal strategy $x_{0}$ is the one that maximizes an expected reward function R(x) for the actions taken with strategy *x*, *D* refers to data from previous games that a player makes use of to estimate R(·), and G represents the player’s maximal possible discernment between strategies. Since the objective is to find the optimal strategy, it is natural to use a truth function
(80)$$f\left(x\right)=1\left(x=x_{0}\right),$$
with the associated set $A=A_{0}=\left\{x_{0}\right\}$ of true worlds corresponding to the upper row of (Equation 1). It follows from Remark 7 that learning and knowledge acquisition are equivalent for game theory whenever (Equation 80) is used. Various algorithms, such as reinforcement learning [53] and sequential sampling models [54,55], could be used by a player in order to generate his beliefs *P* about which strategy is the best.

Many extensions of our work are possible. A first extension would be to generalize the framework of Theorem 1 and Example 1, where data ${\left\{D_{k}=\left(Z_{1},\ldots ,Z_{k}\right)\right\}}_{k=1}^{n}$ are collected sequentially according to a Markov process with increasing state space, without requiring that ${\left\{Z_{l}\right\}}_{l=1}^{n}$ are independent and identically distributed. We will mention two related models for which this framework applies. For both of these models a student’s mastery of *q* skills (which represent knowledge how rather than knowledge that) is of interest. More specifically, $x=(x_{1},\ldots ,x_{q})$ is a binary sequence of length *q*, with $x_{i}=1$ or 0 depending on whether the student has acquired skill *i* or not, whereas $D_{k}$ corresponds to exercises that are given to a student up to time *k*, and the student’s answers to these exercises. It is also known which skills are required to solve each type of exercise. The first model is Bayesian knowledge tracing (BKT) [56], which has recently been analyzed using recurrent neural networks [1]. In BKT, a tutor trains the student to learn the *q* skills, so that the student’s learning profile changes over time. At each time point, the tutor is free to choose the last exercises at time *k* based on previous exercises and what the student learnt up to time k−1. The goal of the tutoring is to reach a state $x_{0}=\left(1,\ldots ,1\right)$ where the student has learned all skills. The most restrictive truth function (Equation 80) monitors whether the student has learned all skills or not, so that $P_{k}\left(A\right)$ is the probability that the student has learnt all skills at time *k*. In view of Remark 7, there is no distinction between learning and knowledge acquisition for such a truth function. A less restrictive truth function $f\left(x\right)=x_{i}$ focuses on whether the student has learnt skill *i* or not, so that $P_{k}\left(A\right)$ is the probability that the student learnt skill *i* at time *k*. The second model—the Bayesian version of Diagnostic Classification Models (DCMs) [57]—can be viewed as an extension of Illustration 1. The purpose of DCMs is not to train the student (as for knowledge tracing), but rather to diagnose the student’s (or respondent’s) current vector $x_{0}=\left(x_{01},\ldots ,x_{0q}\right)$, where $x_{0i}=1$ or 0 if this particular student masters skill (or attribute) *i* or not. The exercises of DCM are usually referred to as items. Assume a truth function (Equation 80); $P_{k}\left(A\right)$ is the probability that the diagnostic test by time *k* has learnt which attributes the student masters. Note in particular that the student’s attribute mastery profile $x_{0}$ is fixed, and it is rather the instructor that learns about $x_{0}$ when the student is being tested on new items.

A second extension would be to consider opinion making and consensus formation [58] for a whole group of *N* agents that are connected according to some social network. In this context, $G_{k}$ represents the maximal amount of discernibility between possible worlds that is possible to achieve after *k* time steps based on external data (available to all agents) and information from other agents (which varies between agents and depends on properties of the social network). It is of interest in this context to study the dynamics of ${\left\{P_{ki}\left(A\right)\right\}}_{i=1}^{N}$ over time, where $P_{ki}\left(A\right)$ represents the belief of agent (or individual) *i* in proposition *S* after *k* time steps. This can be accomplished using a dynamical Bayesian network [59] with *N* nodes i=1,…,N that represent individuals, associating each node *i* with a distribution $P_{ki}$ over the set of possible worlds X, corresponding to the beliefs of agent *i* at time *k*. A particularly interesting example in this vein would be to explore the degree to which social media and social networks can influence learning and knowledge acquisition.

The third possible extension is related to consensus formation, but with a more explicit focus on how *N* decentralized agents make a collective decision. In order to illustrate this, we first describe a related model of cognition in bacterial populations. Marshall [60] has concluded that “the direction of causation in biology is cognition → code → chemicals”. Cognition is observed when there is a discernment and data collection process that either optimizes code or improves the probability of a given chemical outcome. Accordingly, the strong learning process of Definition 3 can be used to model how biological cognition is attained (or at least is expected to be attained). For instance, in quorum sensing, once bacteria reach a critical density, they emit a chemical signal to ascertain the number of neighboring bacteria [61]; when a critical density is reached, the population performs certain functions as a unit (Table 1 of [62] presents several examples of bacterial functions partially controlled by quorum sensing). The proposition under consideration here is *S*: “the function is performed by at least a fraction ε of bacteria”, where ε represents a critical density above which the bacteria act as a unit. The parameter $x=(x_{1},\ldots ,x_{N})$ is a binary sequence reflecting the way in which a population of N=q bacteria acts, so that $x_{i}=1$ if bacterium *i* performs the function, whereas $f\left(x\right)=1\left(x\in A\right)=1\left(\sum_{i}x_{i}\geq \varepsilon N\right)$. For collective decisions, $x_{i}$ rather represents a local decision of agent *i*, whereas f(x) corresponds to the global decision of all agents. Learning about *S* at time k=1,2,⋯ is described by $P_{k}\left(A\right)$, the probability that the population acts as a unit at time *k*. There is a phase transition at time *k* if the probabilities $P_{1}\left(A\right),\ldots ,P_{k-1}\left(A\right)$ of the population acting as a unit are essentially null, whereas $P_{k}\left(A\right)$ becomes positive (and hence $I_{k}^{+}\left(A\right)$ gets large) when discernment ability and data are extended from $\left(G_{k-1},D_{k-1}\right)$ to $\left(G_{k},D_{k}\right)$. This is closely related to the fine-tuning of biological systems [33,35] with *f* being a specificity function and *A* set of highly specified states, and fine-tuning after *k* steps of an algorithm that models the evolution of the system corresponding to $I_{k}^{+}\left(A\right)$ being large. As for the direction of causation from cognition to code, Kolmogorov’s complexity, which measures the complexity of an outcome as the shortest code that produces it, can be used in place of or jointly with active information to measure learning [63].

A fourth theoretical extension is to consider the case |X|=∞. In this case, instead of the (discrete or continuous) uniform distribution given by (Equation 2), it will be necessary to consider more general maximum entropy distributions $P_{0}$, subject to some restrictions, in order to measure learning and knowledge acquisition [20,64,65,66].

A fifth extension is to consider models where the data sets $D_{k}$ are not nested. This is of interest, for instance, in Example 5, when non-nested subsets of confounders are used to predict Adam’s disease status. For such scenarios, it might be preferable to use information-based model selection criteria (such as maximizing AIN) in order to quantify learning [67], rather than sequentially testing various pairs of nested hypotheses by means of
$$\Delta I_{k}^{+}=I_{k}^{+}-I_{k-1}^{+}=\log\frac{P_{k}\left(A\right)}{P_{k-1}\left(A\right)},$$
in order to assess whether learning has occurred in each step *k* (corresponding to strong learning of Definition 3).

A sixth extension would be to compare the proposed Bayesian/frequentist notions of learning and knowledge acquisition, with purely frequentist counterparts. Since learning corresponds to choosing between the two hypotheses in (Equation 18), we may consider a test that rejects the null hypothesis when the log likelihood ratio is small enough, or equivalently, when
(81)$$\Lambda =-2\log\frac{\max_{x\in A}L\left(D|x\right)}{\max_{x\in X}L\left(D|x\right)}\geq t$$
for some appropriately chosen threshold *t*. The frequentist notion of learning is then formulated in terms of error probabilities of type I and II, analogously to (Equation 20), but for the LR-test (Equation 81) rather than the Bayesian/frequentist test (Equation 19) with test statistic AIN, or the purely Bayesian approach that relies on posterior odds (Equation 21). A frequentist version of knowledge acquisition corresponds to using data *D* in order to produce a one-dimensional class of confidence regions CR for $x_{0}$, with a nominal coverage probability of CR that varies. In order to quantify how much knowledge that is acquired, it is possible to use the steepness of a curve that plots the actual coverage probability $P(x_{0}\in \mathrm{CR})$ as a function of the volume |CR|. However, a disadvantage of the frequentist versions of learning and knowledge acquisition is that they do not involve degrees of beliefs, the philosophical starting point of this article. This is related to the critique of frequentist hypothesis testing offered in [68]. Since no prior probabilities are allowed, within a frequentist setting, important notions such as the false report probability (FRP) and true report probability (TRP) are not computable, leading to many non-replicated findings.

A seventh extension is to consider multiple propositions $S_{1},\ldots ,S_{m}$, as in [69,70]. For each possible world x∈X, we let $f:X\to {\left\{0,1\right\}}^{m}$ be a truth function such that $f\left(x\right)=(f_{1}\left(x\right),\ldots ,f_{m}\left(x\right))$, with $f_{i}\left(x\right)=1$ (0) if $S_{i}$ is true (false) in world *x*. It is then of interest to develop a theory of learning and knowledge acquisition of these *m* propositions. To this end, for each $y=\left(y_{1},\ldots ,y_{m}\right)\in {\left\{0,1\right\}}^{m}$, let $A_{y}=\left\{x\in X;\;f\left(x\right)=y\right\}$ refer to the set of worlds for which the truth value of $S_{i}$ is $y_{i}$ for i=1,…,m. Learning is then a matter of determining which $A_{y}$ is true (the one for which $x_{0}\in A_{y}$), whereas justified true beliefs in $S_{1},\ldots ,S_{m}$ amount to finding $x_{0}$ as well. Learning of statements such as $S_{i}\vee S_{j}$ and $S_{i}\wedge S_{i}$ can be addressed using the m=1 theory of this paper, since they correspond to binary-valued truth functions $\tilde{f}\left(x\right)=f_{i}\left(x\right)+f_{j}\left(x\right)-f_{i}\left(x\right)f_{j}\left(x\right)$ and $\tilde{f}\left(x\right)=f_{i}\left(x\right)f_{j}\left(x\right)$, respectively.

## Figures and Tables

**Figure 1 entropy-24-01469-f001:**
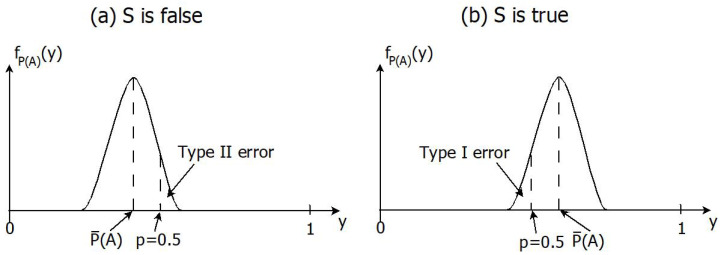
Illustration of the density function $y\to f_{P(A)}\left(y\right)$ of P(A) when the data set $D∼L(\cdot |x_{0})$ varies according to the likelihood of the true world parameter for two scenarios where *S* is either false (**a**) or true (**b**). The threshold of the hypothesis test (Equation 19) is $I^{+}=\log\left[p/P_{0}\left(A\right)\right]$, so that $H_{0}$ is rejected when P(A)≤p=0.5. Note that $\bar{P}\left(A\right)$ is the expected value of each density, whereas the error probabilities of type I and II correspond to the areas under the curves in (**b**) and (**a**) to the left and right of *p*, respectively.

**Figure 2 entropy-24-01469-f002:**
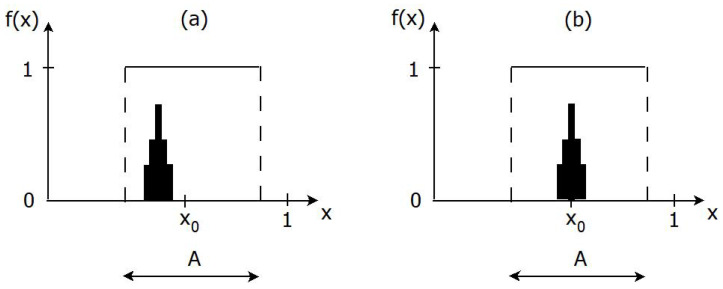
Illustration of the difference between learning and knowledge acquisition for a scenario with a set of worlds X=[0,1] and a statement *S* whose truth function x→f(x) is depicted to the left (**a**) and right (**b**). It is assumed that *S* is true ($x_{0}\in A$), and that the degrees of beliefs $P_{0}$ of an ignorant person correspond to a uniform distribution on X. The filled histograms correspond to the density functions p(x)dx=dP(dx) of two agent’s beliefs. The agent to the left (**a**) has learnt about *S* but not acquired knowledge, since $x_{0}$ does not belong to the support of *P*. The agent to the right has not only learnt about *S*, but also acquired knowledge, since his belief is justified, corresponding to a distribution *P* that is more concentrated around the true world $x_{0}$, compared to the ignorant person. Hence, the JTB condition is satisfied for the agent to the right, but not for the agent to the left.

**Figure 3 entropy-24-01469-f003:**
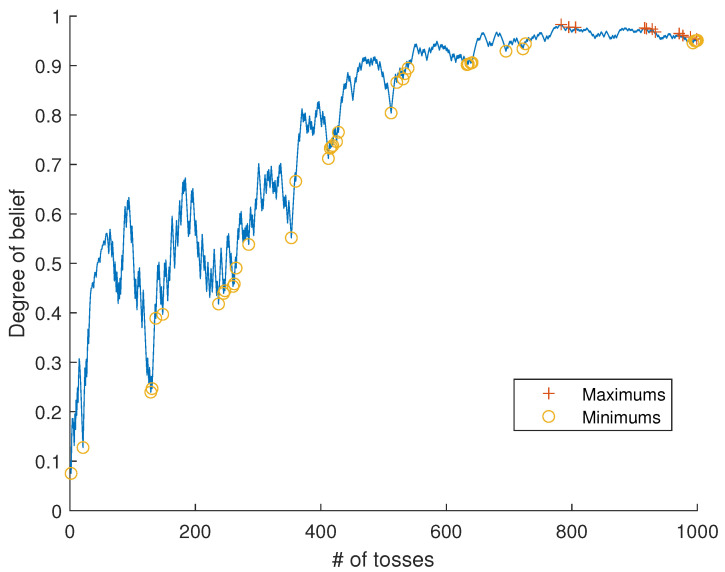
Degree of belief is represented as a function of coin tosses. There is no strong learning because the belief oscillates. However, there is weak learning after a few coin tosses. In particular, when the number of coin tosses is 1000, there is weak learning since $P_{1000}\left(A\right)>P_{0}\left(A\right)$ and $I_{1000}^{+}\left(A\right)>0$.

**Figure 4 entropy-24-01469-f004:**
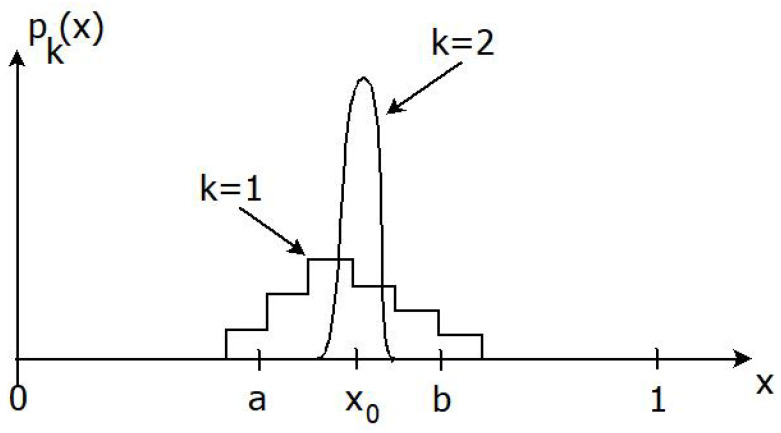
Posterior densities $p_{1}\left(x\right)$ and $p_{2}\left(x\right)$ after one and two steps of the discerment and data collection process of Example 2 when *S* is true ($x_{0}\in \left[a,b\right]$). Since $p_{1}$ is measurable with respect to $G_{1}$, it is piecewise-constant with step length δ. Note that strong learning and knowledge acquisition occurs.

**Figure 5 entropy-24-01469-f005:**
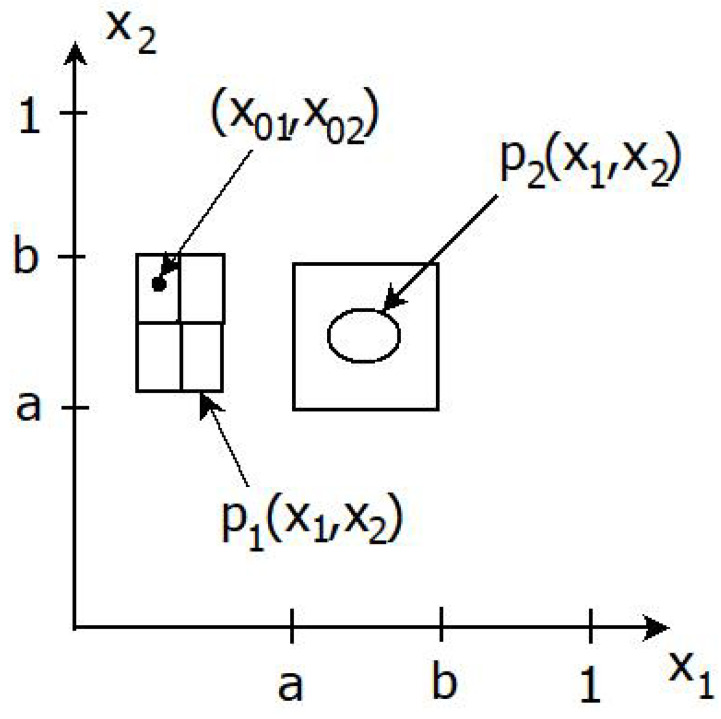
Posterior densities $p_{1}\left(x_{1},x_{2}\right)$ and $p_{2}\left(x_{1},x_{2}\right)$ after one and two steps of data collection for Example 3. Since $x_{01}<a$, it is not possible to have outdoor activities the first day of the camp. The weather forecast density $p_{1}$ is supported and piecewise-constant on the four rectangles with width $\delta_{1}$ and height $\delta_{2}$, corresponding to σ-algebra $G_{1}$. The true temperatures $\left(x_{01},x_{02}\right)$ are within the support of $p_{1}$. On the other hand, the prophet incorrectly predicts that outdoor activities are possible both days; $p_{2}$ is supported on the ellipse. In this case, neither (weak or strong) learning nor knowledge acquisition takes place.

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
