# Peer review of "A Formal Framework for Knowledge Acquisition: Going beyond Machine Learning"

_entropy, 2022, doi:10.3390/e24101469_

Round 1
Reviewer 1 Report
The authors propose a framework based on a kind of model testing to define learning and knowledge. Maybe I miss the point, but the paper is very messy and verbose to finally fall back to testing according to log-likelihood or Bayes factor!
The paper is pretty messy containing many abstruse explanations. The examples from section 8 should be used as running ones along the explanations to help understand all the mathematical modeling. Another option is using an extended teacher example along the text with concrete algebras $\mathcal F$, $\mathcal G$, and $\mathcal G'$. It will help the reader to grasp the framework for JBT-based knowledge definition.
The first difficulty comes from the mix between frequentist and subjective interpretations of probability. Indeed, the knowledge considered as Justified True Belief (JTB) becomes harder to sustain in the Bayesian world: what does "true" means here? JTB condition should be argued.
To help the reader, clear unambiguous notations without repetition are mandatory. For example, S, f, and A model the same notion in section 2. Also, A, for the simplest scenario must deserve a specific notation such as A_0 to avoid confusion in the rest of the paper. Another example of misuse of notation P(A) = P(A|D) is on page 5. Moreover, after that, $D$ is random, see (9). That is pretty sloppy and does bring confusion without any simplification of notation.
Sections 2 and 3 deserve a complete rewrite. A simpler setup such as the one described in Cox's book "The algebra of probable inference" or Jaynes's book "Probability: the logic of sciences"---two books worth mentioning---may be a nice start, and the extension with measurable space and function used later. In Cox/Jaynes's framework, the differences between rational agents' beliefs narrow to the prior choices and the observations.
The set $\mathcal X$ with its algebra $\mathcal F$ is the basic set to settle probabilities. Thus, $g$ in (5) is already a random variable, and you don't need $g(X)$. By the way, the requirement (5) needs deeper explanations. On line 121, what does "physical probability" mean?
What do the authors mean by candidate of $A$ such as in line 136?
On page 5, the sub-algebra $\mathcal G$ is built thanks to a random variable Y that is not observable. What is the need to introduce this variable?
In section 4, AIN is the Bayes factor (in logarithmic scale) or the LLR? What is the novelty? Moreover, why are there equivalence symbols in (18) contrary to the previous discourse that S does not permit to discern the true world $x_0$ (except in the simplest scenario of course)?
Section 5 reduces the knowledge acquisition into the concentration of the agent's posterior as viewed through the "volume" of a closed ball. What a disappointment to get there after reading 8 pages. Other concentration measurements may be considered and must be discussed such as cross-entropy with the Oracle beliefs.
Sections 6 and 7 describe the acquisition process as the sequence of the agent's posterior given the cumulative sequence of data. The corresponding sequence of AINs is bounded or analyzed with standard notions (lim sup/inf, etc). Unfortunately, the aim of both sections is not clear, and I get lost in a list of definitions. What are the objectives of all that stuff?
Section 8 presents five examples, but the proposed framework strength is not emphasized compared to classical Bayesian analysis.
Reviewer 2 Report
I have found the content interesting, and informative. However, I would suggest reformulating the examples to be more practical. In this way, the authors may engage more readers. I strongly advise the authors to represent their solution to the examples via more graphs.
The other point is the structure of sections, please use subsections for the sake of the easiness of following the content.
In some parts, the authors may shorten their text and avoid some very basic formulations (e.g., equations 1, 2, 10,27,...) without losing the main message.
Number the equation on page 5, line 128-129.
Reviewer 3 Report
The paper presents a formal framework for knowledge acquisition. The Authors proposed the framework of learning and knowledge acquisition in a hybrid form between frequentism and Bayesianism. In this paper, they provided an agent’s learning and knowledge acquisition within a mathematical framework of possible worlds. The examples of the applications of the proposed theory were provided. The topic is interesting and the paper is well corresponding to the journal aim and scope.
The paper is quite well structured. The Authors provided the mathematical background of their work. However, there are shortcomings in this paper. In my opinion, the article lacks a schematic/framework in graphic form. This would make the paper easier to read and provide more clarity. Since the Authors of the paper propose a formal framework for knowledge acquisition, it would be worth highlighting this part and presenting it in a more accessible form.
Minor typos:
The paper does not contain paragraphs – please check it according to the template.
Round 2
Reviewer 1 Report
In the last version, the authors took into account all the remarks made in the previous version. Thanks to the running examples and the multiple clarifications of the objectives in each section, the readability of this paper by novice readers is greatly improved.